**Subject Category:**
Biology (whole organism)

palaeontology/taxonomy and systematics

Cambrian, Burgess Shale, bivalved arthropods, inverted swimming, convergence, suspension feeding

**Author for correspondence:**
Alejandro Izquierdo-López
e-mail: ai.lopez@mail.utoronto.ca

# A possible case of inverted lifestyle in a new bivalved arthropod from the Burgess Shale

## Alejandro Izquierdo-López[1] and Jean-Bernard Caron[1,2,3]

[1]Department of Ecology and Evolutionary Biology, University of Toronto, Toronto, Ontario, Canada M5S 3B2
[2]Department of Natural History (Palaeobiology Section), Royal Ontario Museum, 100 Queen's Park, Toronto, Ontario, Canada M5S 2C6
[3]Department of Earth Sciences, University of Toronto, Toronto, Ontario, Canada M5S 3B1

AI-L, 0000-0002-8072-6308

The origin of the arthropod carapace, an enlargement of cephalic tergites, can be traced back to the Cambrian period. However, its disparity and evolution are still not fully understood. Here, we describe a new 'bivalved' arthropod, *Fibulacaris nereidis* gen. et sp. nov., based on 102 specimens from the middle Cambrian (Wuliuan Stage) Burgess Shale, Marble Canyon area in British Columbia's Kootenay National Park, Canada. The laterally compressed carapace covers most of the body. It is fused dorsally and merges anteriorly into a conspicuous postero-ventrally recurved rostrum as long as the carapace and positioned between a pair of backwards-facing pedunculate eyes. The body is homonomous, with approximately 40 weakly sclerotized segments bearing biramous legs with elongate endopods, and ends in a pair of small flap-like caudal rami. *Fibulacaris nereidis* is interpreted as a suspension feeder possibly swimming inverted, in a potential case of convergence with some branchiopods. A Bayesian phylogenetic analysis places it within a group closely related to the extinct Hymenocarina. *Fibulacaris nereidis* is unique in its carapace morphology and overall widens the ecological disparity of Cambrian arthropods and suggests that the evolution of a 'bivalved' carapace and an upside-down lifestyle may have occurred early in stem-group crustaceans.

## 1. Introduction

The study of Cambrian fossils, in particular, those from Burgess Shale and Orsten-type deposits, has been vital to the understanding of the origin and relationships among the main

arthropod groups (chelicerates, trilobites and allied taxa, myriapods, crustaceans) [1–4] and have led to a plethora of different evolutionary scenarios regarding head segments and appendages (e.g. [2–8]). Among cephalic characters, though, the arthropod head shield or carapace has hardly been discussed in character evolution reconstructions or as a part of the cephalic conformation debates. This might be owing to the fact that the embryological origin and development of the head shield or carapace has been subject to different interpretations [9–11]. In addition, disparate patterns of development across different crustacean groups have questioned its homology [12–14]. The terms 'head shield' and 'carapace' have also been defined in different ways by different authors (e.g. [9,10]), even including other cephalic structures in non-crustacean lineages [15]. Here, we define 'carapace' as an enlargement of a cephalic tergite that extends beyond the maxillary segment and covers part or the totality of the body [14]. In recent taxa, the carapace can cover the whole cephalon (e.g. cephalic shield in nauplius larvae), the cephalothorax (e.g. some Decapoda, Notostraca), or even the whole body (e.g. Laevicaudata, Spinicaudata, Ostracoda) [9,10,14]. The carapace may also present a groove, line or suture delimiting two individual elements or valves, and is then termed a 'bivalved' carapace, like in ostracods. The term, though, has been equally used to describe carapaces that extend ventro-laterally on both sides of the body, despite no evidence of hinge line as well, which may reflect a fused condition (e.g. notostracans). Bivalved carapaces with or without a suture line are widely spread across extant eucrustaceans, including the Ostracoda, Notostraca, Cladocera, Laevicaudata, Spinicaudata, Malacostraca, Thecostraca and Branchiura. Whether a carapace, or even a 'bivalved' carapace constitutes a basal trait in Crustacea, has been so far inconclusive, though. Despite this, crustaceans with reduced or absent carapaces (i.e. tanaidaceans, isopods, anostracans) appeared relatively late in crustacean evolution (e.g. *Lepidocaris*) [16–18] and some arthropods with carapaces from the upper Cambrian (e.g. *Rehbachiella*) have already postulated as basal branchiopods [19], indicating an early origin of the trait.

The arthropod carapace, in fact, was present in multiple groups during the Cambrian, including a number of putative stem-group crustaceans or even pancrustaceans [20] which include the Phosphatocopina [21–23] and Bradoriida [24,25], both of which are fully enclosed in bivalved carapaces (see also: [26,27]). In addition, the Hymenocarina (i.e. *Tokummia, Waptia, Branchiocaris*) [3] possesses a bivalved cephalothoracic carapace, as well as mandibular and maxillary adaptations, which brings them close to stem mandibulates [28], suggesting that the bivalved condition predates a pancrustacean origin. Other Cambrian bivalved arthropods include groups such as the Isoxydae (e.g. *Isoxys, Surusicaris*) [29,30] potentially close to radiodonts [31], as well as taxa of uncertain affinities (e.g. *Nereocaris, Loricicaris, Plenocaris*) [32,33], suggesting that Cambrian bivalved arthropods are most probably a polyphyletic group [28,33].

Here, *Fibulacaris nereidis* gen et sp. nov., a new arthropod from the middle Cambrian (Wuliuan Stage) Burgess Shale (British Columbia, Canada), with a seemingly unique carapace morphology is described. The carapace is 'bivalved' in the sense that it extends along both sides ventrally, so that a right and left side of the carapace can be differentiated, but no hinge line is present. Furthermore, it extends into a strikingly long rostrum, which bends ventrally so that it lies parallel to the body, an uncommon carapace trait, which also offers an interesting functional challenge. Furthermore, its potential ecologic niche and feeding mode, as well as its taxonomic affinities are inferred, which stress the ecological and taxonomical disparity of this group.

# 2. Material and methods

## 2.1. Collection and observations

We examined a total of 102 specimens (electronic supplementary material files), collected from talus slopes or *in situ* from two localities in the Burgess Shale fossil localities in Kootenay National Park (British Columbia, Canada): 11 specimens were obtained from the locality of Marble Canyon [34] from two expeditions in 2012 and 2014. A total of 91 specimens come from different expeditions along Mount Whymper and Tokumm Creek, in 2014 and 2016 and 2018, respectively [35]. All specimens are deposited at the Royal Ontario Museum Invertebrate Palaeontology (ROMIP) Collections. Some specimens were further prepared with an air scribe in order to remove any matrix covering anatomical features. Specimens were observed using a stereomicroscope with cross-polarized filters and photographed dry and wet under direct or cross-polarized lighting. Measurements were made on specimens with full carapaces ($n = 50$).

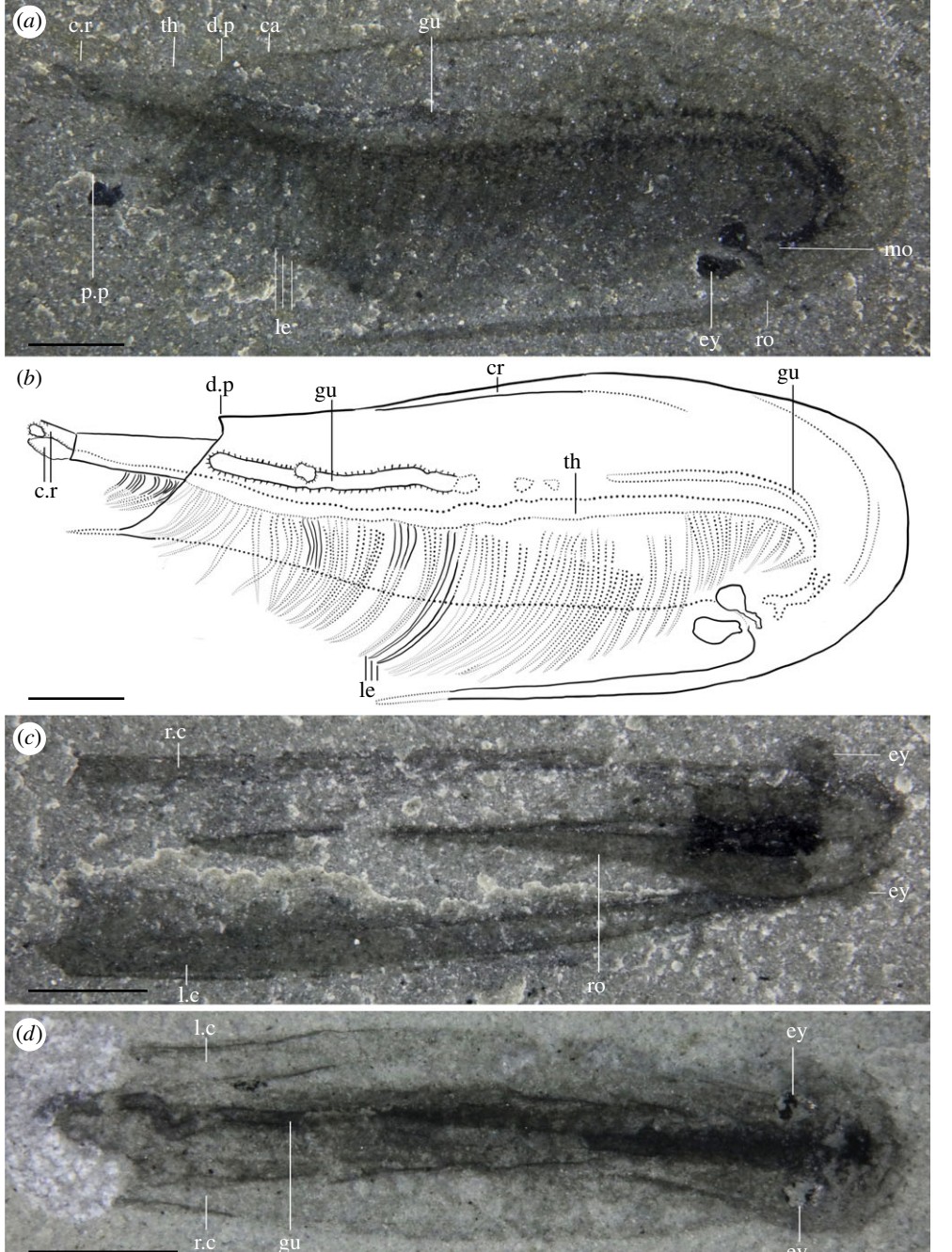

**Figure 1.** General anatomy. (*a*) ROMIP 65380 (dry), lateral view, holotype. (*b*) Drawing of (*a*). (*c*) ROMIP 65461 (wet), ventral view showing rostrum. (*d*) ROMIP 65460 (wet), dorsal view showing gut trace. ca, carapace; cr, crest; c.r, caudal rami; d.p, dorsal process; ey, eye; gu, gut; le, legs; l.r, left side of the carapace; mo, mouth; p.p, postero-lateral process; r.c, right side of the carapace; ro, rostrum; th, thorax. Scale bars: 2 mm (*a–d*).

## 2.2. Taphonomy

Most specimens are preserved flat in lateral view, with only minor differences in their angle of burial (e.g. figures 1*a*, 4*f* and 5*d*; although see electronic supplementary material, figure S2) and only a few specimens preserved in dorsal or ventral view (figures 1*c,d* and 3). Taken together, this suggests that most specimens were preserved along their largest and flattest plane of symmetry. Left and right side of the carapace were narrowly separated along their ventral and posterior margins, as evidenced by specimens which preserve both sides of the carapace on top of each other (e.g. figures 2*e*, 4*a* and 5*i*) and by specimens preserved in ventral or dorsal views (figures 1*c,d* and 3). Carapaces vary in their length-width ratios (e.g. figures 1*a*; 2*e*; 4*a* and 5*j,k*; electronic supplementary material, figure S5),

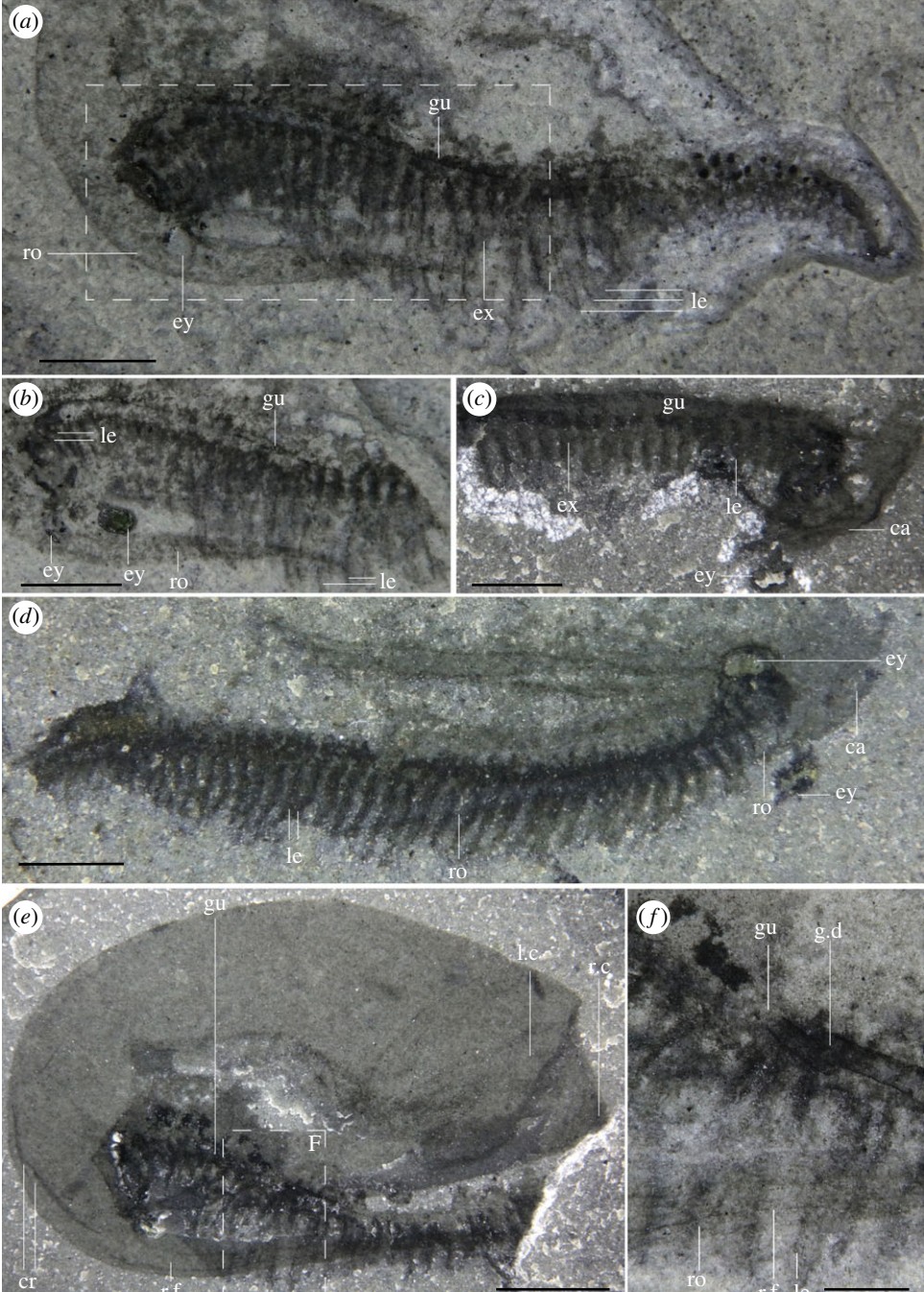

**Figure 2.** Anatomy of legs, the cephalic region and displacement of the gut pellets. (*a*) ROMIP 65397 (wet), lateral view. (*b*) ROMIP 65397 counterpart (wet), close-up of the cephalo-thorax showing details of the legs (image flipped horizontally). (*c*) ROMIP 65458 (dry), close-up of the cephalon-thorax showing probably exopods, lateral view. (*d*) ROMIP 65462 (wet), specimen showing legs along most of the length of the body, lateral view. (*e*) ROMIP 65400 (dry), lateral view. (*f*) ROMIP 65400 (wet), close-up of the gut. ca, carapace; cr, crest; ex, exopods; ey, eye; gu, gut; g.d, gut displacement; le, legs; l.c, left side carapace; r.c, right side carapace; ro, rostrum; r.f, rostrum furrow. Scale bars: 4 mm (*e*); 2 mm (*a*–*d*); 1 mm (*f*).

which can be generally attributed to different preservation angles, as well as various degrees of folding and compression (e.g. figure 5*d,j*). There is no evidence of the presence of different morphotypes although the number of measurable specimens is too low to run any statistical tests (electronic supplementary material, figure S5). The carapace extends into a rostrum that recurves ventrally, parallel to the ventral margin of the carapace (e.g. figures 1*a*; 2*e*; 5*a,k*). In rare occasions, some specimens show a slight separation of the rostrum away from the ventral margin of the carapace and a higher degree of curvature (most extreme in two specimens: figures 4*a* and 5*d*), which we interpret

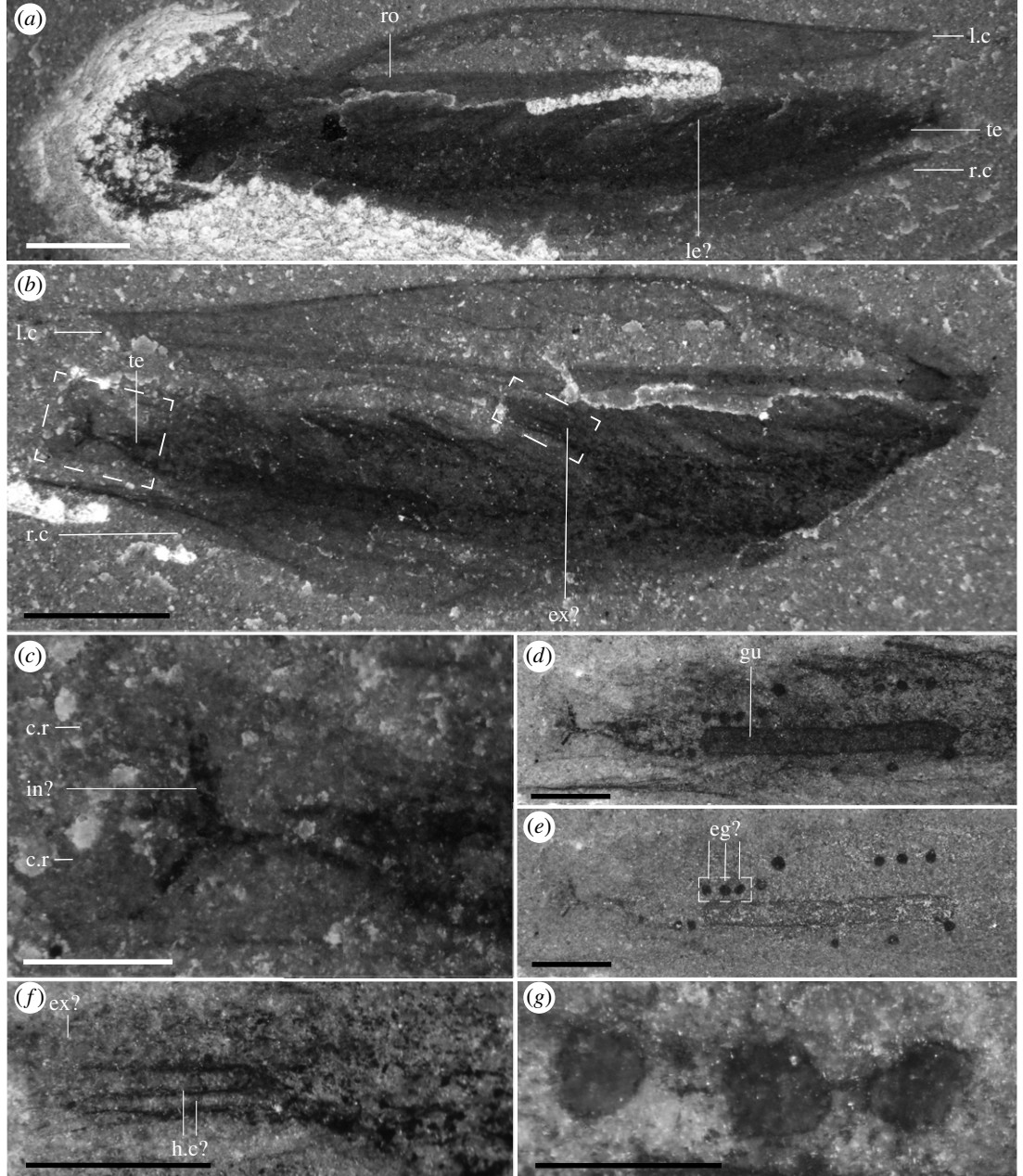

**Figure 3.** Ventral views of ROMIP 65359. (*a*) Part (dry). (*b*) Counterpart (dry). (*c*) Close up of the telson on the counterpart (wet), showing indeterminate tissues. (*d,e*) Close up of the thoracic-telson region on the counterpart (wet) using polarizer (*d*) and no polarizer (*e*) to show the location of the presumed eggs. (*f*) Close-up of a putative exopod on the counterpart (see *b*) showing possible preservation of haemolymph cavities. (*g*) Close-up of possible eggs on the counterpart. c.r, caudal rami; eg?, egg (possible); ex?, exopod (possible); gu, gut; h.c?, haemolymph cavities (possible); in?, indeterminate feature; le?, legs (possible); l.c, left side carapace; r.c, right side carapace; ro, rostrum; te, telson. Scale bars: 2 mm (*a,b*); 1 mm (*d,e*); 0.5 mm (*c,f*); 0.25 mm (*g*).

to be taphonomic in origin. The rostrum is generally straight, tapers to a thin point when fully exposed, and like the rest of the carapace, does not show evidence of breakage.

The body is generally poorly preserved, and in many cases, only traces of the gut and eyes remain, which suggest that the body was weakly sclerotized and decayed away faster than the carapace (figures 4*a* and 5*d,k*). Despite this, multiple specimens are fully articulated (electronic supplementary material, S1), and preserve other soft-tissues, including legs. Eyes are preserved as conspicuous dark reflective films, often retaining three-dimensionality (figures 1*a,d* and 5*a,d*). Legs are usually poorly preserved, and anatomical details, such as the boundary between podomeres are not recovered. Multiple specimens show three-dimensionally preserved rod-like structures, black, that run across the centre-dorsal part of the body along the gut (e.g. figures 2*f*, 3*d* and 4*a,d,g*). These pellet-like structures

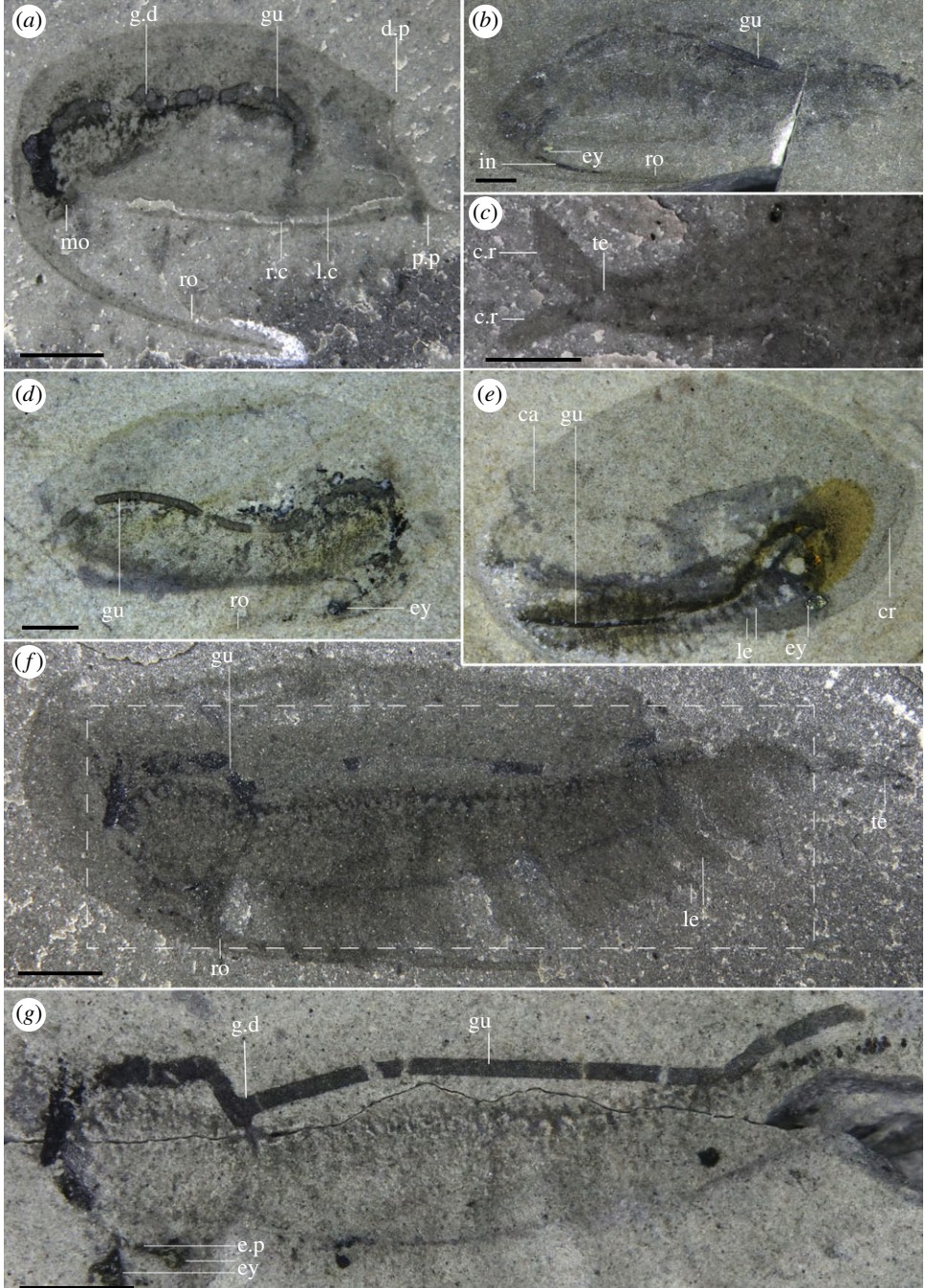

**Figure 4.** General anatomy and gut preservation. (*a*) ROMIP 65369 (dry), lateral view showing gut and taphonomic displacement of the rostrum. (*b*) ROMIP 65363 (wet), lateral view showing gut and indeterminate internal tissues within rostrum (see also the electronic supplementary material, figure S1*b*,*c*). (*c*) ROMIP 65457 (dry), ventral view, detail of the telson (see also the electronic supplementary material, figure S2). (*d*) ROMIP 65390 (wet), lateral view showing gut. (*e*) ROMIP 65388 (wet), lateral view showing gut. (*f*) ROMIP 65401 (dry), lateral view showing gut and limbs. (*g*) ROMIP 65401 (wet), composite images of part and counterpart to show the full extent of the gut. ca, carapace; cr, crest; c.r, caudal rami; d.p, dorsal process; e.p, eye peduncle; ey, eye; g.d, gut displacement; gu, gut; in, indeterminate tissue; le, legs; l.c, left side of the carapace; mo, mouth; p.p, postero-lateral process; r.c, right side of the carapace; ro, rostrum; te, telson. Scale bars: 2 mm (*a*,*b*,*d*,*e*,*f*,*g*); 1 mm (*c*).

appear generally rigid, moulding the entire diameter of the gut (e.g. figure 4*b*) and are present following the sinusoidal outline of the body (figure 4*d*) from head to anus or just part of it, and sometimes away from both mouth and anus (e.g. figure 4*b*). Furthermore, these structures appear discontinuous, twisted or broken *in situ* in multiple specimens (e.g. figures 2*f* and 4*a*). In some cases, different sections appear to be preserved highly displaced from the lumen, even in specimens showing no indication that the body

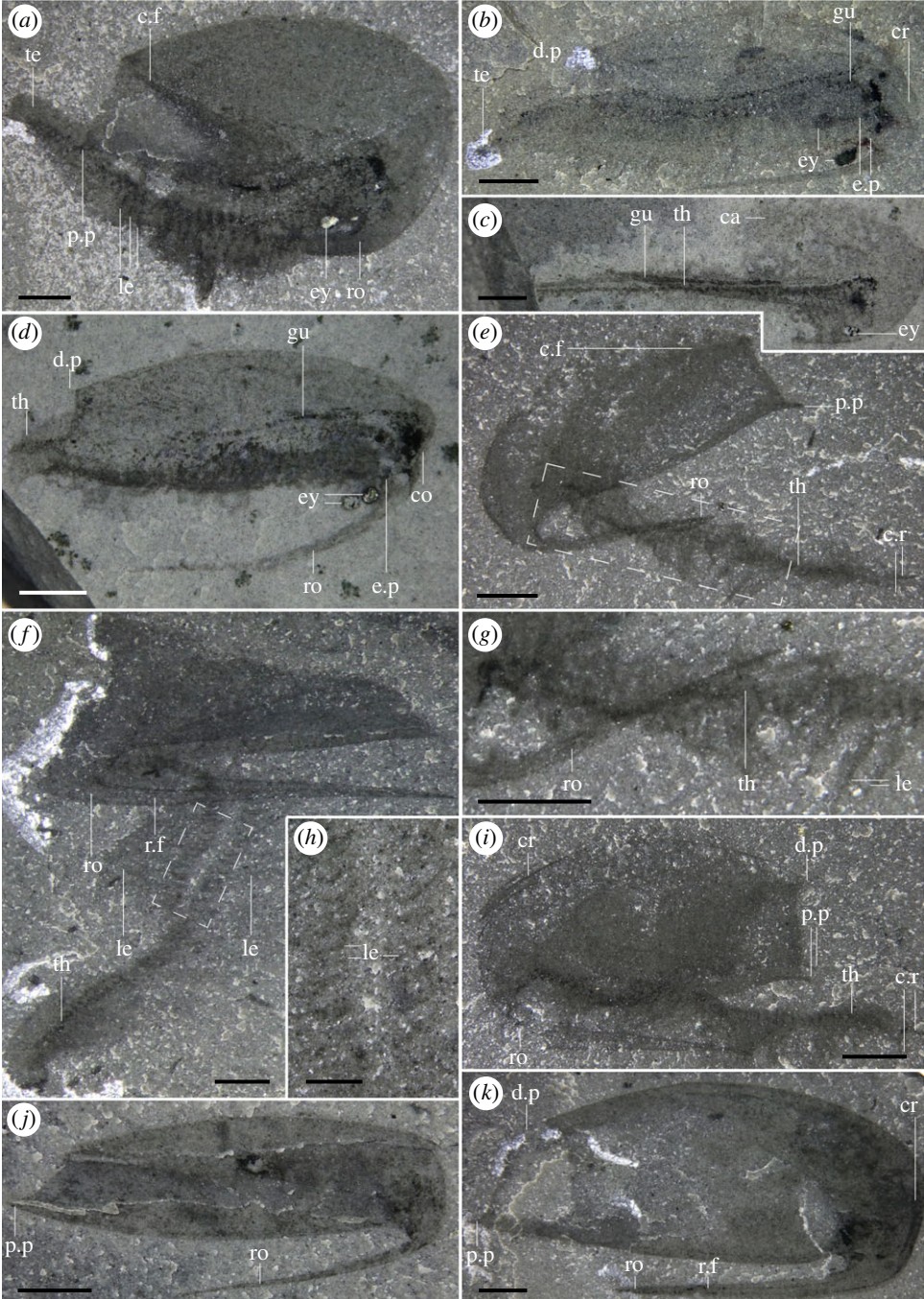

**Figure 5.** Specimens with different preservation states. (*a*) ROMIP 65395 (dry), lateral view, well-preserved specimen but with the body rotated ventrally and partially overlapping the rostrum. (*b*) ROMIP 65355 (dry), lateral view, weathered specimen but still showing evidence of legs and internal tissues. (*c*) ROMIP 65383 (wet), lateral view with internal tissues mostly preserved. (*d*) ROMIP 65385 (wet), oblique specimen showing the exceptional rotation of rostrum interpreted to be taphonomic in nature. (*e,g*) ROMIP 65379 (dry), potential moult, full view (*e*) and close up of the rostrum and thoracic region (*g*). (*f,h*) ROMIP 65382 (dry), potential moult, full view (*f*), and close-up of the legs (*h*). (*i*) ROMIP 65401 (dry), lateral view. (*j*) ROMIP 65368 (wet), oblique view, specimen showing no evidence of soft tissues. (*k*) ROMIP 64511 (dry), lateral view, specimen showing no evidence of soft tissues. ca, carapace; c.f, carapace folds; co, compression; cr, crest; c.r, caudal rami; d.p, dorsal process; e.p, eye peduncle; ey, eye; gu, gut; le, legs; mo, mouth; p.p, postero-lateral process; ro, rostrum; r.f, rostrum furrow; te, telson; th, thorax. Scale bars: 2 mm (*a,b,c,d,e,f,g,i,j,k*); 1 mm (*h*).

has been displaced from its carapace (e.g. figure 4*a,g*), suggesting that the breakages followed an early mineralization of the gut track. Although elemental maps showed no differences in the composition of these structures and the rest of the matrix, like other Burgess Shale fossils the gut was initially

probably phosphatized and then replaced during late-phase diagenesis by aluminosilicate minerals [36] (see also *Mode of Life*). A separate black element on the rostrum of one specimen is rich in minerals containing calcium and phosphorus (figure 4*b*; electronic supplementary material, figure S1) which is probably the remnants of the primary mineral phase.

Specimens with carapaces upturned and tilted forward are interpreted to represent exuviae (figure 5*e*, *f,i*). Moulting behaviour in these specimens is supported by the fact that the rest of the body seems flimsy and twisted and does not show evidence of eyes and gut (which normally preserve well, see above). The body remains attached to the carapace anteriorly (figure 5*e,f,i*), implying that the connection between body and carapace was strong. In most cases, specimens were found isolated, but the discovery of a number of clusters suggests a potential gregarious behaviour (e.g. electronic supplementary material, figure S2; ROMIP 65455, ROMIP 65456).

## 2.3. Phylogenetic methods

A phylogenetic analysis was performed based on the datasets published in Vannier *et al.* [28] and Aria & Caron [3], from which a total of 74 adult taxa and three larval taxa were used that have a carapace (Phosphatocopina, *Bredocaris*, *Rehbachiella*). To this dataset, 13 new or recoded adult species were added, including *F. nereidis*, for a total of 90 taxa (electronic supplementary material, S2). Four new and revised characters related to the telson (electronic supplementary material, S3) were added to 209 selected characters from that matrix, for a total of 213 characters. All characters were discrete and multistate and were considered unweighted and unordered for the phylogenetic analysis. Characters without a clear support from the fossil material were treated as uncertainties.

The tree search was performed through a Bayesian analysis using MRBAYES v. 3.2.6 [37]. We used a Markov k model with gamma distribution as used previously [28]. The analysis ran for 20 000 000 generations, with two runs and four chains. Records of trees and parameters were printed every 1000th generation. A 20% burn-in was subsequently applied. A backbone constraint was applied mostly for extant taxa, based on molecular phylogenies [38–40] (electronic supplementary material, S4). Differently to the previous analysis, molecular clocks were only applied at the root of the tree. *Opabinia* was chosen as the outgroup.

# 3. Results

## 3.1. Systematic palaeontology

*Superphylum* Panarthropoda Nielsen, 1995
 *Phylum* Arthropoda von Siebold, 1848 (see [41])
 Subphylum Mandibulata Snodgrass, 1938
 Genus *Fibulacaris* gen. nov.
 Genus: urn:lsid:zoobank.org: B75A9D2C-3925-42A3-9AD7-37800FC667BF
 Etymology: from the ancient roman brooch 'fibula' and the ancient Greek word 'caris' for crustacean.
 *Diagnosis*: Euarthropod with the following characters: bivalved carapace extending anteriorly into a single postero-ventrally recurved rostrum, parallel or subparallel to the body, reaching a maximum of the total length of the carapace. Posteriorly, the carapace terminates in a small dorsal process, and each valve terminates in a stout spine pointing posteriorly. The carapace exceeds the body height mid-way along its length, leaving a wide space dorsally. The body has *ca* 40 homonomous segments with the cephalic region curving ventrally.
 Type and only known species: *Fibulacaris nereidis* sp. nov.
 Species: urn:lsid:zoobank.org:B75A9D2C-3925-42A3-9AD7-37800FC667BF
 *Etymology*: from Nereids, sea nymphs in Greek mythology.
 Diagnosis: as for the species, by reason of monotypy.
 *Holotype*: ROMIP 65380
 *Paratypes*: ROMIP 65397, ROMIP 65400
 *Additional material*: 99 specimens figured, referenced or measured in the article (electronic supplementary material, S1).
 *Occurrence*: Middle Cambrian, Wuliuan Stage, Burgess Shale. A total of 11 specimens were collected from Marble Canyon [34] and 91 specimens from Tokumm Creek (including Mount Whymper) [35], both occurring on top of the thick Stephen Formation in Kootenay National Park, British Columbia, Canada.

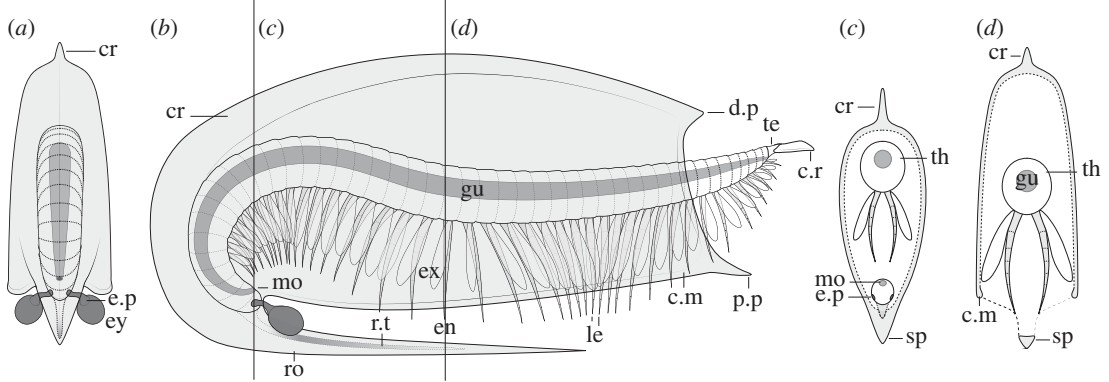

**Figure 6.** Diagrammatic reconstruction of *Fibulacaris nereidis*. (*a*) Frontal view. (*b*) Lateral view. (*c*) Cross-section across the cephalic region. (*d*) Cross-section across the thorax. cr, crest; c.m, carapace margin reinforcements; c.r, caudal rami; d.p, dorsal process; en, endopod; ex, exopod; e.p, eye peduncle; ey, eye; gu, gut; mo, mouth; p.p, posterior-lateral process; ro, rostrum; r.t, rostrum unidentified tissue; te, telson; th, thorax.

*Measurements*: average length: 12.74 mm (s.d.: 0.36), average height: 4.75 mm (s.d.: 0.19) ($n = 50$). Maximum length: 20.0 mm (ROMIP 65363). Maximum height: 10.0 mm (ROMIP 65386). Electronic supplementary material, figure S5.

## 3.2. Description

### 3.2.1. Carapace

The 'bivalved' carapace is thin with slightly reinforced margins (e.g. figure 5*e*). The carapace is slender and laterally compressed as seen in ventral or dorsal views (figures 1*c,d* and 3) and soft parts (legs, gut) are visible without removal of the carapace (figures 1*a* and 2*e*). This contrasts with similar Burgess Shale 'bivalved' arthropods which have thicker carapaces and a more ovoid cross-section (e.g. *Perspicaris*, *Plenocaris*, *Canadaspis*). The carapace is ovoid in lateral view with the anterior margin semicircular and the posterior margin oblique to slightly concave (e.g. figure 1*a,b*). The dorsal margin is slightly convex, while the ventral margin is generally straight to slightly concave (e.g. figure 4*a*). The dorsal margin of the carapace terminates posteriorly into a single small process (dorsal process) (e.g. figures 4*a* and 5*d*) while ventrally, each side of the carapace ends in a strong postero-lateral spine-shaped process, directed backwards (e.g. figures 1*a*; 2*e* and 3*a*). The two sides of the carapace are fused dorsally and frontally and there is no evidence of a hinge line. In displaced moults, the body appears detached frontally from the carapace, indicating that the carapace probably originates from a cephalic segment or early thoracic segment (figure 5*e,f*). The carapace extends frontally into a single medially positioned and postero-ventrally recurved rostrum, parallel (e.g. figures 1*a* and 5*a*) or subparallel (e.g. figures 4*a* and 5*d*) to the ventral margin of the carapace. The carapace covers the head tightly and most of the body more loosely, leaving a large dorsal space (figure 6*c,d*) and less than a third of the thoracic segments and the telson uncovered (e.g. figures 1*a*; 3*a* and 5*a*). The rostrum extends more than half of the carapace length in most specimens (e.g. figure 1*a*) or, more rarely, the entire length of the carapace and tapers to a thin point. The rostrum is usually straight (see *Taphonomy*) and sturdy and is probably hollow in cross-section as demonstrated by the presence of phosphatized minerals (figure 4*b*; electronic supplementary material, figure S1*b*)

Several specimens exhibit folds that run across the carapace, either frontally (e.g. figure 2*e*), dorsally (figures 4*a*; 5*b,i,k*), obliquely (figure 5*a,b,e*) or on the rostrum (e.g. figures 2*e,f* and 5*f,g,k*). Dorsal and frontal folds may suggest the presence of a crest, running from the dorsal part and connecting to the rostrum (figure 6) which may give rise to these folds post-compression. Folds on the rostrum may also be the result of a slight furrow running longitudinally. Oblique deformations may suggest that the two sides of the carapace could be bent slightly outwards (as seen in ventral view, figure 3*a,b*), and may also suggest that the dorsal part of the carapace may have been partly empty, allowing for the posterior part of the body to fully retract inside it (figure 6*c,d*). Specimens that show legs positioned at different heights with respect to the carapace (e.g. figures 2*a,e*, 4*f* and 5*a*), or specimens

that appear curled (figure 4a) similarly support the idea that the posterior part of the body may have been able to move dorsoventrally into the carapace cavity to some extent.

### 3.2.2. Head

The head is completely covered by the carapace. The anterior part of the body curves ventrally, directing the head and the mouth opening posteriorly (e.g. figures 1a and 4a), partially shielded by the inner side of the rostrum. No clear differentiation exists between the cephalic and thoracic region, but this could be owing to the small size of the specimens and relatively poor preservation of the legs. Most specimens preserve two compound eyes directed ventrally and facing posteriorly (e.g. figures 1a and 4g,h) which seem ovoid and are as big as 5% of the total body length (e.g. figure 5a,b,d). The eyes appear to project behind the base of the rostrum from two stalks seemingly as long as the eyes themselves (figures 1a; 4g and 5d).

### 3.2.3. Thorax

The thorax is multisegmented and is composed of up to approximately 40 weakly sclerotized segments poorly preserved in most cases with an unknown number of segments belonging to the head itself (e.g. figure 2d and ROMIP 65396). Segments are homonomous, generally identical, though tapering distally in height closer to the front and telson. Traces of legs are seemingly present in every segment from limb-bearing cephalic or thoracic segments up to the telson (e.g. figures 1a,b and 2d), implying that the abdominal region is either very short or non-existent.

### 3.2.4. Legs

Each segment bears at least one pair of legs (e.g. figures 2f and 5h), to a total of approximately 40 pairs of legs (figure 2d). Legs seem to originate just behind the presumptive mouth area (figure 2b) and appear in each segment up to the telson, decreasing in length posteriorly (e.g. figures 1a, 4f and 5a,b). Legs close to the head also appear to be reduced in length relative to mid-thoracic legs (figure 2a). Overall, legs are flimsy and elongated, and the distal parts normally protrude out of the carapace (figures 1a; 2a and 5b), even protruding beyond the rostrum, in cases where the body appears rotated ventrally (figures 4f and 5a).

Most legs preserve as dark elongate sheaths, considered endopods, and can be individually inferred in multiple specimens (e.g. figures 2d,f and 5g,h). These are weakly arthrodized, and again, owing to their small size, lack the detail required to allow for a good estimate of their shape, exact number of podomeres and endites. Nevertheless, the total number of podomeres in F. nereidis is assumed to be seven, which is thought to represent the ground pattern in euarthropods [42]. The presence of exopods is inferred based on dark traces which originate near the base of the legs (figures 2a–d; 4e and 5a) like in other, similarly preserved arthropods (e.g. Nereocaris) [33]. Although not well-preserved, these traces suggest that the exopods may have had a paddle-like shape and a length slightly shorter than half the total length of the endopod (figures 2c and 5a). One specimen shows what might appear to be the basal part of an exopod with possibly haemolymph-filled cavities (figure 3f) reminiscent of those described in Surusicaris [30].

### 3.2.5. Gut

The gut itself is not well preserved, but the presence of dark three-dimensional rod-like elements demarcates the position of the mouth (e.g. figures 4a and 5k). These elements indicate that the gut was straight with no evidence of gut diverticulae or additional glands (see also §2.2 Taphonomy above and §4.3 Mode of Life below).

### 3.2.6. Telson

The last trunk segment appears after the anus (e.g. figures 3d and 4c), and is thereby considered a telson. The telson is overall mostly undifferentiated from other segments (figures 1a and 4c), and bears on its posterior margin two small fan-shaped caudal rami (e.g. figures 1a; 3b,c and 4c) which appear laterally folded in some specimens (figure 5e,i).

### 3.2.7. Other traits

One specimen shows small spherical, three-dimensional elements (*ca* 0.1 mm) across the posterior segments of the trunk, especially apparent in ventral view (figure 3*d,e,g*) that are not found anywhere else on the matrix. These structures appear grouped but unconnected to each other. They also appear unconnected to the gut, as no connection can be inferred. This potentially rules out the possibility that these structures are auxiliary digestive glands, as seen in other Burgess Shale arthropods (e.g. *Leanchoilia*) [43]. Because of their size and position, these structures may represent eggs, which have also been recorded in other 'bivalved' arthropods from the Burgess Shale (e.g. *Waptia*) [44]. However, this interpretation is tentative, and further studies and additional specimens would be required to assess the nature of these structures.

### 3.2.8. Phylogenetic results

The Bayesian phylogenetic analysis (figure 7; electronic supplementary material, figure S3) recovers *F. nereidis* gen. et sp. nov. in a monophyletic group containing other 'bivalved' arthropods with highly multi-segmented bodies: *Loricicaris*, *Nereocaris exilis*, *Nereocaris briggsi*, but which also includes *Clypecaris* and *Perspicaris*, which have lower numbers of segments. Posterior probabilities for most of the nodes in this group are regarded as low (less than 0.6). This group is retrieved closely related to the Hymenocarina [3], and possibly Fuxianhuiida [45].

Other analyses without the additional characters (electronic supplementary material, figure S4) added to Vannier *et al.* [28] recover a generally similar structure. The additional characters, mainly 'subdivisions of the caudal rami' separate *F. nereidis* (not segmented) from *N. exilis* (tri-segmented), otherwise recovered as sister taxa (electronic supplementary material, figure S3), although with general low support values in all analyses. The position of the Hymenocarina as a stem mandibulate aligns with previous analyses [3], but contrasts with the analysis of Vannier *et al.* [28], in which the Hymenocarina were found as stem-group Pancrustacea.

## 4. Discussion

### 4.1. Taxonomic affinities and notes on cephalic structures

Resolving the precise relationship of *F. nereidis* is not straightforward, as many details of cephalic and limb morphology are not well-preserved (as emphasized by the low node support in our phylogenetic analysis). However, *F. nereidis* is placed close to other Cambrian 'bivalved' arthropods (figure 7; electronic supplementary material, figure S3), including *Nereocaris*. A likely relationship of *Nereocaris* with or close to Hymenocarina contrasts with previous analysis that placed it close to basal euarthropods [32,33]. *Fibulacaris nereidis* has a long, homonomous and highly multi-segmented thorax without a thoracic-abdomen differentiation, which is shared with the Hymenocarina taxa *Tokummia* [3] and *Branchiocaris* [46]. On the other hand, the long undifferentiated and weakly cuticularized legs of *F. nereidis* resemble those in *Nereocaris*, *Loricicaris* and possibly *Perspicaris* [2,20]. Whether or not these are homologous to the legs in hymenocarines (e.g. *Tokummia* and *Branchiocaris*), which have a clearer structure of seven podomeres and terminal claws would need further testing. The carapace of *F. nereidis* does not possess a hinge line, as in the hymenocarina *Waptia fieldensis* [28], and is frontally fused, as in *Erjiecaris minuscule*, of uncertain affinity [47]. The longitudinal crest in *F. nereidis* could potentially be homologous to the dorsal keel of *N. exilis* and similar to that in *Jugatacaris agilis* [48]. The postero-ventrally recurved rostrum is also reminiscent to the two antero-ventral processes in *N. exilis* [32], but these neither reach the same size nor the same degree of curvature. Other frontal processes are equally present in less closely related (figure 7) arthropods (e.g. *Tuzoia* [49] and *Isoxys* [50]). In fact, several species of *Isoxys* (e.g. *Isoxys volucris*, *Isoxys longissimus*, *Isoxys paradoxus*) bear elongated anterior and posterior processes [31] similar in length or even longer than the rostrum in *F. nereidis*. In addition, spine-shaped rostrums are also commonly found across extant malacostracan larvae, suggesting this may represent a labile, convergent character [51]. However, these structures are normally projected outwards, either frontally or laterally and thus, the position of the rostrum in *F. nereidis* seems to be, *a priori* unique in extinct and extant taxa, although analogous structures may exist (see §4.3 *Mode of Life*).

More generally, the results of the phylogenetic analysis may also hint to a basal origin of a 'bivalved' carapace in mandibulates, although this would be difficult to ascertain, given that myriapods and fuxianhuiids present 'head shields', but not a carapace (bivalved or not) covering part of their thoracic segments. Considering the basal position of Isoxyiidae, the bivalved carapace could have had an early

origin in arthropod evolution, especially if other non-crustaceomorph carapace-structures (carapaces in Hurdiidae and Artiopoda) were found to be homologous, a question that still remains. Regarding other cephalic structures, *F. nereidis* together with the closely related taxon *Nereocaris* (and also *Odaraia* which is retrieved as a basal Hymenocarina, figure 7) are also unique in that these taxa do not seem to have antennae or potential antennal homologues (i.e. chelicerae, great appendages [52,53]), and neither seem to have mandibles. Antennae are not only present in most mandibulates, but also in most of the Euarthropoda and the presence of mandibles is a key apomorphy in mandibulates and, although controversial for fuxianhuiids [45], it is overall accepted in Hymenocarina taxa [3,28,46,54,55]. Antennae and antennulae appear reduced in multiple branchiopod (s.l) groups (e.g. cladocerans, spinicaudatans, laevicaudatans, anostracans), and if not taphonomic, *F. nereidis* could have similarly present reduced, not recognizable antennae, in a potential case of evolutionary convergence. Furthermore, it is possible that species closely-related to *F. nereidis* could have lost the antennae (in *Nereocaris*) or mandibles (in *Clypecaris, Perspicaris, Nereocaris*) as a result of a change to a deposit or suspension-feeding ecology, as inferred for these taxa. The absence of these features, though, could arguably be also taphonomic in nature, owing to the lack of well-preserved material or the small size in the case of *F. nereidis*. *Waptia*, which is known from hundreds of specimens comparatively only shows the mandibles clearly in a few specimens [28]. Future revisions of other Cambrian taxa such as the larger *Perspicaris recondita* [20], including new material from the Burgess Shale might help refine our understanding of limb and cephalic morphology and the phylogenetic position of some of these taxa within or outside Hymenocarina.

## 4.2. Carapace functional morphology

The basic function of the carapace was most probably protection. The postero-lateral processes in *F. nereidis* bear some resemblance to the posterior spines in some cladocerans (e.g. Daphniidae) and, as in these taxa [56], these processes could have had an anti-predatory function. The long posteriorly directed ventral rostrum of *F. nereidis* could have had a similar role, protecting the ventral part of the animal during swimming. Some cladocerans (e.g. *Bosmina, Neothrix, Parophryoxus, Peracantha*) [57,58] show anterior posteriorly directed ventral projections, which can also be highly elongated (see [59]). These projections, though, are not part of the carapace itself, but instead they represent elongations of the head shield, partly embedding two parallel antennulae to different degrees in different species. These structures are known to act as levers (e.g. *Neothrix*) [58], to penetrate into soft deposits (e.g. *Parophryoxus*) [58], as defensive structures (e.g. *Bosmina*) [59] and can bear terminal sensillia, implying a sensory function (e.g. *Macrotrhix, Parophryoxus*) [58]. Because it is part of the carapace, the rostrum in *F. nereidis* most likely did not carry sensilla and certainly did not act as a lever, but it could hypothetically have been used to penetrate into soft deposits or stir the sediment, acting as a plough, a common strategy in deposit and suspension feeders [56,58,60].

Carapace structures such as rostrums or spines are also known to increase drag and stability during swimming [31,56,61–63], increasing swimming speed, directionality, reducing energy expenditure and/or enabling specific motions [14,63,64]. Multiple telephinid trilobites possess equally elongated, ventrally directed facial spines [65] and species such as the remopleurid *Hypodicranotus* have posteriorly directed, elongated hypostomes, both structures possibly analogous of the rostrum in *F. nereidis*. In the case of *Hypodicranotus*, this structure mainly helps stabilizing and reducing drag [66]. Whether the carapace morphology of *F. nereidis* could have had a hydrodynamic role though, is difficult to ascertain in the absence of hydrodynamic modelling.

Besides external currents, carapace morphology influences internal currents too. In multiple extant bivalved species (e.g. Cladocera, Ostracoda, Laevicaudata, Spinicaudata) the carapace, together with movement from the antennae, antennulae or trunk legs, creates a current necessary for both filtration and suspension feeding [67]. As the cephalic region of *F. nereidis* is covered by the carapace, food must have been acquired and transported to the mouth from the posterior to the anterior part of the body. Therefore, the most probable mode of feeding would have been through the creation of currents using limb metachronal movement, as in other 'bivalved' arthropods [55]. This type of movement is usually tied to phyllopod or exopod movement, and sometimes aided by sticky secretions that gather the suspended material (in some Ostracoda and Cladocera) [57]. This mode of feeding may be further supported by a convergent shape of the frontal part of the body, ventrally curved both in both *F. nereidis* and Cladocera.

## 4.3. Mode of life

The mode of life of *F. nereidis* is here inferred from the overall morphology of the preserved parts of the animal, including its carapace, thorax, legs and gut. As stated, the carapace could have aided in

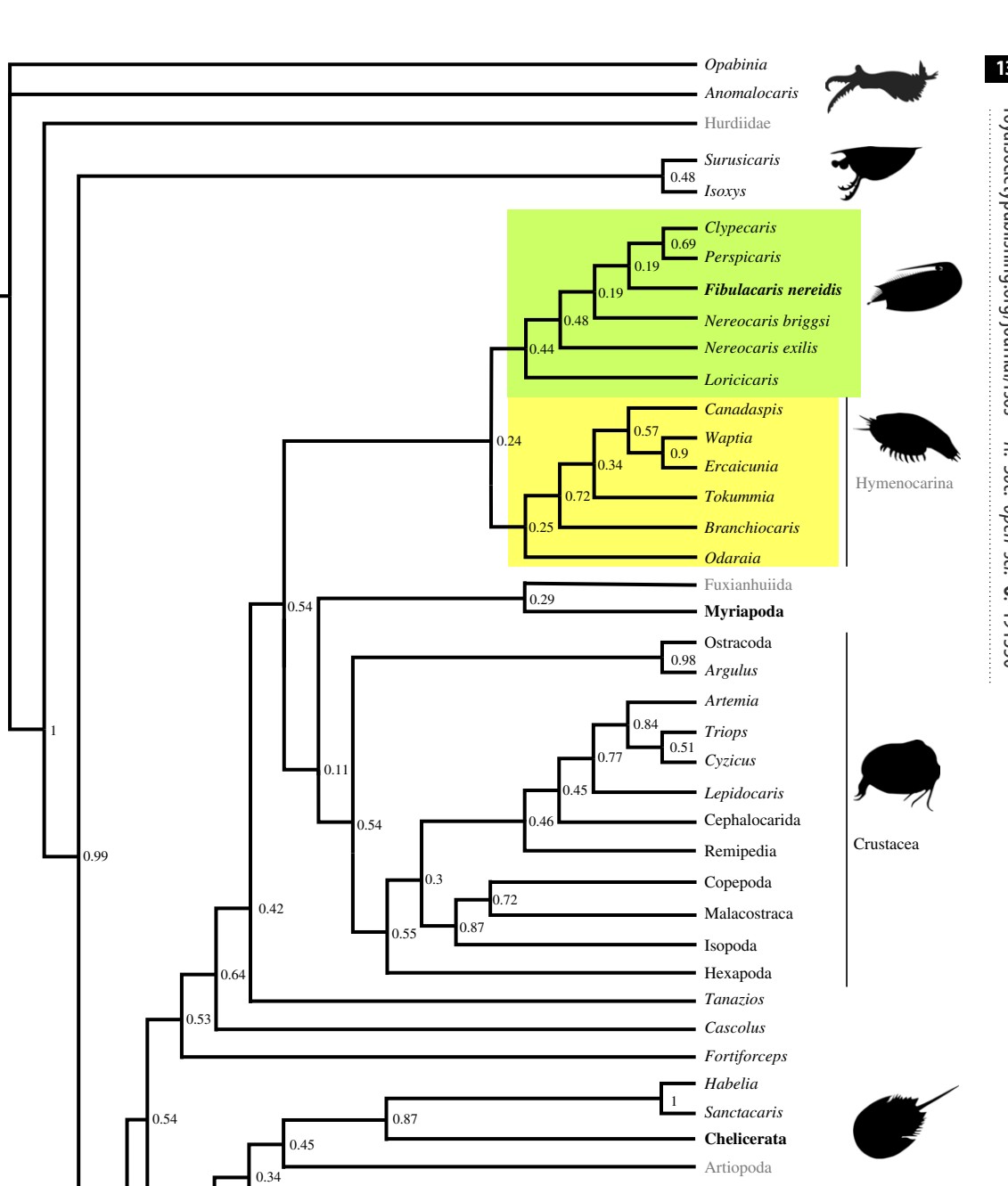

**Figure 7.** Interpretative cladogram based on a consensus tree from a Bayesian analysis using a Markov k model on a morphological dataset with 90 taxa and 213 characters (see the electronic supplementary material, figure S1). Numbers next to nodes are posterior probabilities. The yellow box indicates the new monophyletic group to which *Fibulacaris* belongs to. The green box highlights the group Hymenocarina.

the creation of filter-inducing current, thus suggesting a filter or suspension-feeding habit, which is further supported by the elongated, multisegmented thorax without specialized legs [68]. The gut is long and simple, without evidence of adjacent glands, which is also associated with both suspension and deposit feeding, as in both cases it is necessary for particles to remain inside the gut long enough to perform particle selection [69]. In some extant taxa, in fact, the animal resuspends buried particles and subsequently collects them, combining deposit and suspension feeding [69]. If regarded as a deposit feeder, though, the legs in *F. nereidis* may have been too short and flimsy to have been used for walking or crawling and its rostrum would represent a challenge, as it would

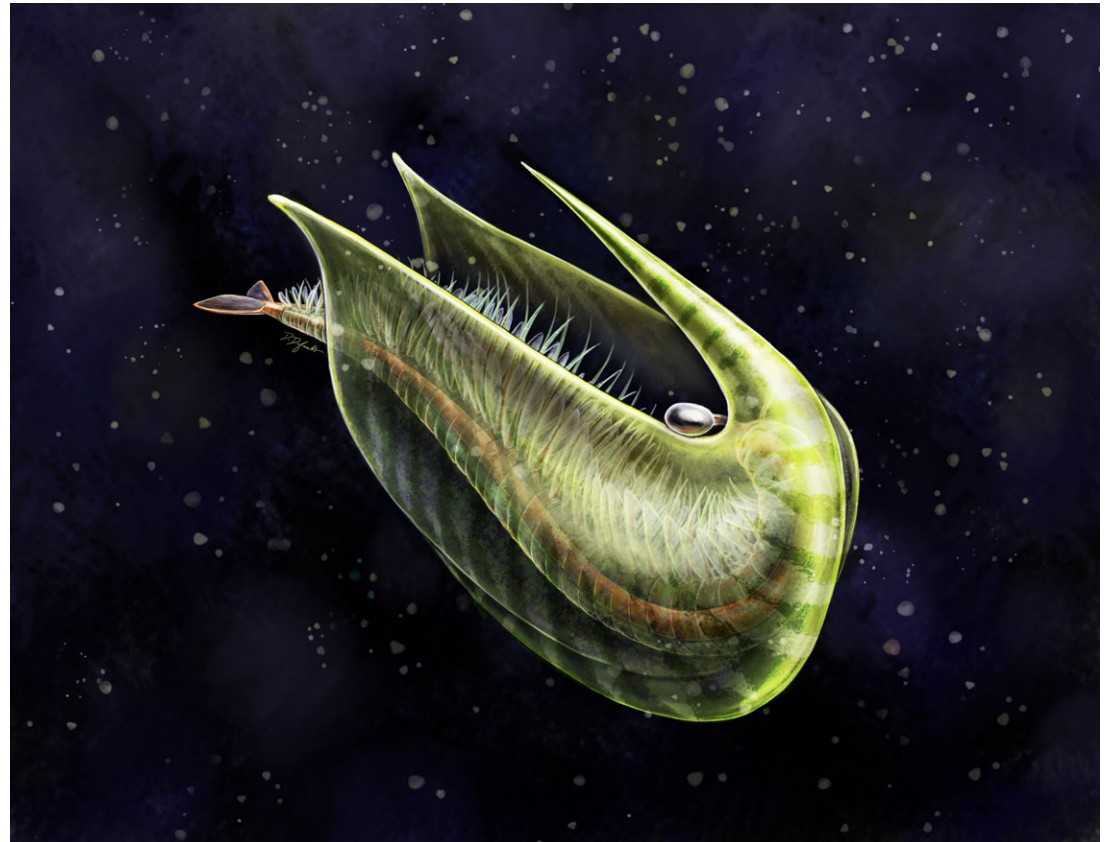

**Figure 8.** Artistic reconstruction of *Fibulacaris nereidis* while swimming upside-down (inverted swimming). Courtesy of Danielle Dufault © Royal Ontario Museum.

require to be totally or partially buried. Choosing between these different scenarios also depends on the interpretation of the three-dimensional structures preserved in the gut. Gut preservation is common in Burgess Shale arthropods, in many cases preserved as three-dimensional structures [68,70]. Three-dimensional guts resembling the surrounding matrix have been traditionally regarded as evidence for deposit feeding in other 'bivalved' arthropods from the Burgess Shale (e.g. *N. briggsi*, *N. exilis*, *Plenocaris plena*) [32,33,71]. In *F. nereidis*, the origin of the solid gut structures is unlikely the result of content ingested while the animal was trapped in muddy turbid flows [72], because in most cases, the three-dimensional structures appear far from the mouth or anus. Decay experiments in the suspension feeder *Artemia* (Anostraca) have shown that bacteria-induced authigenic mineralization occurs early, in particular (although not restricted to) when gut contents are present, while the rest of the body, including muscles supporting the gut, decays more readily [73], a process which could have similarly occurred in a suspension feeder *F. nereidis* (see §2.2 *Taphonomy*).

If regarded as a benthic suspension feeder, *F. nereidis* most likely swam upside-down (inverted swimming), similar to anostracans (figure 8). This behaviour is supported by its posteriorly directed eyes and mouth, as well as the rostral spine protecting the ventral side of the body and the legs. Inverse natation is rare in arthropods, and is mainly limited to the aforementioned Anostraca, the Xiphosura [74], some Cladocera [57], and in some conditions, Notostraca [75] and has similarly been inferred in the Cambrian arthropods *Odaraia* [55] and *Sarotrocercus* [76]. Inverted swimming has been suggested to aid bringing the centre of gravity lower than in the usual orientation, thus increasing balance and stability, and avoiding the clogging of the mouth in deposit and suspension feeders, further facilitating the collection of suspended particles [60]. In *Odaraia* (electronic supplementary material, figure S1*a*), another possible suspension feeder, the hydrodynamic properties of the carapace would have assisted in lift only when swimming inverted [55]. Likewise, this could have been the case in *Fibulacaris*, although modelling of the hydrodynamic properties of the carapace would be needed to strengthen this case. Traits such as homonomous segments and undifferentiated legs enclosed in a carapace have been argued to be especially suitable for inverted swimming [60] and are also found in notostracans and cladocerans. In the case of *Odaraia* and *Sarotrocercus*, the presence of

phyllopodous legs, was associated with filter feeding and respiration and used as evidence in their reconstruction [55,76]. However, not all crustaceans with phyllopodous legs use inverted swimming (e.g. Leptostraca, Spinicaudata, Laevicaudata), and neither do all filter feeders. Nevertheless, based on the overall morphology, the ventral rostrum and a potential suspension-feeding habitat, *F. nereidis* may have swum inverted, probably while feeding, whereas movement across the water column could have been mostly abdominal and restricted to escape reactions, as inferred in morphologically similar species [32].

# 5. Conclusion

*Fibulacaris nereidis* contributes to the increasing morphological, functional, ecologic and taxonomic diversity of bivalved arthropods known from the Cambrian period. The shape of the carapace, with its single posteriorly directed ventral rostrum, appears to be morphologically unique not only among Cambrian and other fossil species but similarly rare across extant crustaceans or other arthropods. The carapace, including the rostrum, most probably had a protective role, but as in other extant arthropods, could have contributed in swimming performance and the creation of feeding currents. Furthermore, *F. nereidis* represents a potential case of inverted swimming (figure 8), rare across arthropods and analogous to that observed in anostracans and some cladocerans. These findings highlight the importance of the carapace morphology in palaeo-ecological reconstructions and show that the arthropod carapace was already both a morphologically and functionally diverse character in the Cambrian period. The phylogenetic analysis (figure 7) reveals a potential new group of mandibulate deposit and suspension feeders with homonomous legs and segments, with some lacking certain mandibulate characters, such as antennae or mandibles, which may be related to an adaptation to this ecological niche and further illustrate a case of convergent evolution with some branchiopod (s.l.) taxa. Furthermore, these results suggest that the bivalved carapace could have been a basal trait for all Mandibulata or may even have had an earlier origin if this and the bivalved carapace of the Isoxyiidae were found to be homologous. However, homologies between arthropod carapaces, bivalved or not, and structures such as radiodont shields [77,78], non-crustaceomorph univalved carapaces (e.g. *Burgessia*, *Naraoia*) and head shields (e.g. fuxianhuiids, habeliids, nauplius) are still poorly understood, which hinders any comprehensive evolutionary analysis on this trait. Besides, new data and morphological revisions on key bivalved arthropods could reshape the present phylogenetic analyses. Nonetheless, Cambrian bivalved arthropods certainly show a high ecological and taxonomic disparity, that is increasingly contributing to the understanding of the evolution of early arthropods and the Cambrian period as a whole.

Data accessibility. Datasets related to this article have been uploaded as part of the electronic supplementary material.
Authors' contributions. Observed and analysed the material: A.I.-L., J.-B.C. Prepared material: J.-B.C. Photographed material: J.-B.C. Compiled bibliography: A.I.-L. Compiled morphological matrix and performed phylogenetic analysis: A.I.-L. Created figures: J.-B.C., A.I.-L. Wrote the manuscript: A.I.-L., J.-B.C. Both authors gave their final approval for submission.
Competing interests. We declare no competing interests.
Funding. This research was undertaken as part of A.I.-L.'s doctoral studies and was supported by doctoral fellowships from the University of Toronto (Department of Ecology and Evolutionary Biology), a Natural Sciences and Engineering Research Council of Canada Discovery grant to J.-B.C. grant no. (341944), the Dorothy Strelsin Foundation (ROM) and Becas Posgrado en América del Norte y Asia-Pacífico (Fundación La Caixa, Spain). Fieldwork activities were supported by the Royal Ontario Museum (Research and collection grants, Natural History field work grants), the Polk Milstein Family, the National Geographic Society (grant no. 9475-14 to J.-B.C.), the Swedish Research Council (to Michael Streng), the National Science Foundation (NSF-EAR-1554897) and Pomona College (to Robert R. Gaines).
Acknowledgements. We thank an anonymous reviewer, as well as Pierre Gueriau for their time and useful comments, which have greatly improved our manuscript. We thank the staff of the palaeobiology section at the Royal Ontario Museum, in particular, Danielle Dufault for the technical drawings and reconstruction and Maryam Akrami and Peter Fenton for their assistance in the collections. We also thank Sara Scharf for editorial suggestions. A.I.-L. thanks Cédric Aria for discussions of the phylogenetic analysis, as well as Joseph Moysiuk, Karma Nanglu and Justin Moon for their constructive comments and general support. We thank Parks Canada, in particular, Todd Keith, for facilitating fieldwork activities. Fossils for this study were collected by Royal Ontario Museum field parties under several Parks Canada Research and Collections permits to J.-B.C. (YNP2012-12054, KOONIP 2014-16317; YNP-2016-21639; KOONP-2018-28179). This is Royal Ontario Museum Burgess Shale project number 86.

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
