## [Reviewer comments · Royal Society Open Science]

Review History

RSOS-191350.R0 (Original submission)

Review form: Reviewer 1

Is the manuscript scientifically sound in its present form?

No

Are the interpretations and conclusions justified by the results?

No

Is the language acceptable?

Yes

Do you have any ethical concerns with this paper?

No

Have you any concerns about statistical analyses in this paper?

No

Recommendation?

Major revision is needed (please make suggestions in comments)

Comments to the Author(s)

Title. The title should not contain the interpretation/speculation about its swimming posture up front and instead should reflect the fact that the bulk of this paper is a description of a new arthropod with a remarkable antero-ventral feature. The present versions comes across as attention-seeking.

Abstract. The first sentence can be deleted. Oblong in which view, plan view or side view? "Fused anteriorly" means what? The captions use legs, not limbs. "Bristle-like limbs" means what exactly? It is unlikely that a biramous arthropod leg would have the shape of a bristle. The abstract should not use we and our. It is not necessary to include the phylogenetic analysis since it does not show anything special. "Retrieves" and "recovers" may both be used in cladistics, but considering the average reader, they are awkward verbs for the suggested position in the tree.

Introduction. This is much too long and over-referenced. It should set the stage relatively briefly for the paper which is mostly descriptive, rather than being a long story about the nature and evolution of the carapace; some of this would better belong in the discussion, but better focused. Some carapaces are fused. Please avoid the all-inclusive "our". The text cannot speak for everyone.

Taphonomy. Thin and "very thin" cannot be distinguished. This section mixes observations with interpretations. This is best avoided. Why does preservation with the animals lying on their sides indicate they "were normally turned sideways during transport"? What is the evidence for transport rather than the carcasses just lying or landing on their sides upon death? "Close bedding planes"? But the valves were fused were they not? The "ventral spine" seems like a long and recurved (or antero-ventrally curved) tapering rostrum – perhaps calling it a ventral spine is the wrong term. That it is not "normally showing breakage" indicates that sometimes it does – is this true, or should this be reworded? The "slight separation of the spine away from the ventral margin," could this be just a morphological variation in the curvature rather than deformation? It seems the spine is stout and its base wide as it merges into the carapace, so breakage there might seem improbable. How straight is "very straight"? (In other words, please avoid using indefinable adjectives.) "Strikingly black" is another. The degree of upturning cannot be "strong". What explanations other than moulting could there be for this attitude? So, what is the interpretation of the black rodlike feature? Does "carbon, calcium and phosphate elements" mean these as elements or something else? Calcium and phosphorus do not occur as elements in these shales.

The phylogenetic tree is not needed here. It does not differ all that much from Figure 3 of Legg et al. (2012), Figure 6 of Legg and Vannier (2013), and Vannier et al. (2018) in the relationships between the taxa of interest in the colored boxes, and the rest of the tree for the Pancrustacea etc. is not particularly relevant to the new form. The additional species incorporated in it do not add any real insight to the appreciation of *Fibulacaris*.

Systematic. Panarthropoda and Arthropoda should not be italicized. Shorten the genus etymology. The diagnosis for the genus is too long: it is more like a mini-description. In any case, since the genus is monospecific, here best is to put "as for the species, by reason of monotypy." Line 14 need the genus epithet. Since there are 47 specimens, the complete list should be provided. In the opinion of this reviewer, it is not necessary to select two paratypes out of the whole collection. The diagnosis should stand alone (and be brief). This is because the species in the tangible object whereas the genus is a concept, not a thing. Line 24 gives a different number of specimens than in the Material section. Is there a reason why it is named after nereids?

Carapace. This could be a little shorter, and in places better worded. It is probably not necessary to refer to all the figures in detail if they are listed under the species name. This is unclear: “Both valves connect dorsally and extend slightly into a small dorsal process”. Where is this, considering the ventral portion of the carapace becomes the downward curving spine? The posteriorly projecting processes on the carapace are best not termed “ventral processes” but posterolateral processes or spines as one would with phyllocarids. The carapace hinge is fused so it is not quite right to use valves in the plural. This is confusing: “anterior part is rotated ventro-posteriorly into a single conspicuous spine” because a carapace does not rotate; later the abstract states the body is rotated, so this is confusing. Elsewhere the body is rotated. The dorsal crest or keel may be vaguely shown in Figure 4A but it does not seem obvious in the other images, and line 17 is vague about this. Maybe there is no crest and the reconstruction is wrong in this regard. An axial crest might seem somewhat unlikely, even if the valves are fused.

Structure—why structure and why not include this in the section above? “Reinforce the idea” should be reworded. “Cephalic fold” is also an unconventional term for grooves or furrows on the carapace. Carapace fold. Spine folds. The description of possible burial-induced wrinkles should use clearer terms and not imply that they are a carina or furrow. Under Head there is no mention of folds. Perhaps all the things labelled folds may not be folds, whatever is meant by the term. All this needs to be clarified. Interpretation is creeping into this section and it should be removed and put with the discussion.

Thorax. What is the evidence that *Nereocaris* is closely related, and how close is that? This is discussion and does not belong in the species description. Same with interpretation of the gut in the next paragraph, and referring to the caudal rami as weakly sclerotized.

The different subsections should be partly amalgamated.

Discussion. The valve–thorax shape is common in many taxa. “Could bear some resemblance” is awkward. So is “clear defensive function”. “Rostral spine” appears here for the first time—it is a better term than ventral spine and “ventro-posterior spine” which comes later in this section. Again “highly elongated” and “highly multisegmented” are descriptions that have no agreed-upon scale; “closely related” is not appropriate. The carapace is not truly bivalved if it is fused, and presumably it was, in order for a rostrum to taper into a long curving spine. Some comparison with the trilobites that have a long spine projecting from the anterior border would be worthwhile. Also perhaps to trilobite hypostomes; some bear posteriorly directed projections.

Mode of Life. This section can be tightened. This reviewer is not certain that line 35 is correct, that the 3D nature of the gut is as critical. Gut contents to some extent (i.e. organic matter versus ingested sediment). How could this happen: “postmortem mud-infilled ingestions of sediment”? The upside-down swimming is worth suggesting but it is basically a suggestion or speculation, and “argue” (line 34) is not appropriate given the rationale in the paragraph. Carolinites is not much of an analog and the genal spines are posteriorly directed; the free cheek is angled ventrally. McCormick and Fortey did not make a case that it swam upside-down, so line 16 is just a speculation made in passing here, and does not support the case. Other possible interpretations apart from being inverted?

This discussion should be more concise and more logically organized. At present it jumps around. It needs to be careful in separating sensible interpretation from outright speculation, with a reduction of the latter.

Bibliography (should be References). The journal states it is not fussy about format and they will put the entries into their own house style, but this list makes little attempt to follow it and comes

across as somewhat sloppy. What does “internet” refer to? In any case, there are too many references for a paper like this. The number can be reduced by maybe a third. For example, in the Introduction basic statements are supported by up to half a dozen citations.

Figure captions. It seems that a bolded title is not needed. “Highly preserved” should be well preserved. Close-up of, not on. The captions should be more explanatory, not just a list of specimen numbers. This would help the reader immeasurably.

Images. These are good. They should all be in black and white to get the tones uniform.

Figure 1. Probably just B is needed as the dry version A does not show any additional feature or the structures more clearly. C is dry? Is C needed? Which portion of the gut in A/B? Since these are all from the same specimen it is not necessary to repeat the number.

Figure 2. It might be an idea to flip B and D so that it can be compared with A. In B, how certain is the right eye? It could be labelled in C. Is D dry? Gut displacement could have a different name, like flexure. Valves?

Figure 3. What is the black blob at the anterior end? Possible is better than putative. The three features in F do not look like cavities. Setae? The black objects might be eggs but the text refers to them as “traces”. D and E are not both wet? The location of F should be noted (with an arrow). Is it necessary to label a feature “indeterminate”?

Figure 4. Lateral views. B is superfluous, and maybe C is not useful. “Part and counterpart overlap” means what? Which part is shown in H?

Figure 6. The letters are too small. Need C and D. The photographic images did not specifically show the dorsal ridge or crest. The caption is too brief. Maybe a ventral view too.

Figure 7 should have been completed before submission.

Figure 8. Fibulacaris is considered closely related to Nereocaris but on the basis of what evidence? The caption is too brief. It should also explain the colors etc. What group is Malacrustacea? This is not a taxon in common usage.

Figure 9 should just show Fibulacaris. Page 12 refers to Marrella but this is not relevant and the comparison vague.

Review form: Reviewer 2 (Pierre Gueriau)

Is the manuscript scientifically sound in its present form?

Yes

Are the interpretations and conclusions justified by the results?

Yes

Is the language acceptable?

Yes

Do you have any ethical concerns with this paper?

No

Have you any concerns about statistical analyses in this paper?

No

Recommendation?

Accept with minor revision (please list in comments)

Comments to the Author(s)

Manuscript RSOS-191350 reports on a new bivalved arthropod from the renewed Cambrian Burgess Shale, which is clearly different from any other fossils described so far and as such definitely deserves to be published. The fossil description is neat and rather well illustrated, and the text is overall well organized and well written, yet the manuscript (particularly the references and the conclusion) is unfinished, and there are (in my opinion) some issues, listed below, that need to be addressed before the manuscript can be considered for publication.

Major comments:

(Note that I refer to page and line numbers provided in the pdf I had access to online, which do not correspond to real line numbers.)

- While the authors are very careful in their conclusions in the main text, making common use of 'potential.ly', 'possible.bly', etc. their title and abstract are not, and I would suggest them to keep careful therein as well. I am here particularly referring to the title and the use of 'confirms' in the last sentence of the abstract, while the authors are way more careful in their discussion.

- In the title, as well as in many places along the text, the authors refer to the long ventral spine of their fossil as a 'ventro-posterior spine' (even 'posterior-ventral spine' p.12 l.43), but this spine is NOT ventro-posterior because it is actually positioned at the anterior end of the animal but is directed towards the posterior. I would rather use 'posteriorly-directed ventral spine'.

- There is a big mess with references numbering (a bit detailed below), which absolutely needs to be carefully checked before publication.

- As the fossils are quite small, the authors state that some anatomical features such as the limbs are poorly preserved or at least hard to observe. This is even more the case in some of the photographs, and I deplore the lack of drawings to help the reader identifying some of the anatomical features described. Similarly, (except for fig. 9) close-ups are not associated with boxes in the corresponding large-scale photographs and as such are often hard to locate...

- Authors for the taxa are not given. Please add them when you first mention a taxon.

- The phylogeny is probably the main issue of this paper. Besides the fact that Aria & Caron's phylogenies (on which this work is based) are far from making consensus among the community, I noticed the following issues with the way the analysis was performed and the data are presented:

(1) The authors only present a simplified version of a consensus tree from a Bayesian analysis (plus one tree as suppl. fig.), but totally discard parsimony analyses, which in my opinion should be also produced at least for comparison.

(2) The 'simplified version of a consensus tree' is actually NOT a simplified version, as it shows relationships different than in the complete tree presented as suppl. figure; for instance, radiodonts appear monophyletic in Fig. 8, but are absolutely not in the complete tree is suppl. Fig., which is another serious issue...

(3) Each new paper using Aria & Caron's phylogeny show important topological changes among the trees, questioning its stability. Here, there is also notable differences with the tree published

by Vannier et al. 2018, on which the present analysis is largely based (in particular the position of Hymenocarina and Myriapoda). This should be discussed much more than the couple of sentences p.9 l.41-48. And the new topology should also be compared to those in both papers published by Aria & Caron in 2017 (your refs [4, 5]).

(4) The 'four new characters' added (p.5 l.23) should be described in the main text, and the effect of their addition at least discussed (showing trees with and without these characters, at least in Suppl. Material, would be very useful). Note that they are not much discussed in Suppl. Material either.

(5) The backbone constraint used (p.5 l-3) can only be found in the matrix nexus file, but nowhere else in the main text or suppl. material.

- Fig. 3A, the long posteriorly-directed ventral spine seems segmented/annulated, is this due to the quality of the photograph in the pdf version, or could it be a real feature?

- And a totally open question: Considering that Fibulacaris specimens are quite small (maximum length: 2 cm) compared to most other bivalved arthropods from the Burgess Shale, and considering also that conspicuous spines are common in larval/juvenile crustacean stages, couldn't Fibulacaris represent a larval/juvenile form of a larger bivalved arthropod found in the same layers? Of course, the putative eggs (an unlikely interpretation in my opinion) would make them mature individuals, but this constitutes a weak (and very unlikely) argument to rule out the possibility that they were larval/juvenile forms.

Additional comments / corrections:

(Note that I refer to page and line numbers provided in the pdf I had access to online, which do not correspond to real line numbers.)

- p.2 l.27, the convergence with trilobites that appears here in the abstract is only superficially discussed (in a single sentence) in the text, so the comparison appears here very weak.

- p.2 l.33, the interesting conclusion/implication that the bivalved carapace may have been a basal trait for Mandibulata could be included into the abstract.

- p.3 l.7, please illustrate the different type of carapace covering by giving names of modern organisms showing such features.

- p.3 l.14, please indicate a date for the Anostraca, and rather refer to a fossil than to a molecular clock that we know is hard to trust... Actually, the Cambrian Rehbachiella (stem-group Branchiopoda) had a carapace, and the oldest-known fossil anostracans (*Lepidocaris rhyniensis* and *Haltinaias serrata*) are from the Devonian (Early and Late Devonian, respectively).

- p.3 l.36, here starts the mess with references numbering. I am quite sure that ref. [50] should be 48.

- p.3 l.37, I am quite sure that ref. [51] should be 49.

- p.3 l.39, I am quite sure that ref. [52] should be 50.

Check all the references numbering from this point, as the problem persists almost all along the rest of the text.

- p.3 l.39, 'as well AS feeding modes'

- p.4 l.16-17, '(ROMIP) Collections'

- p.4 l.53, Fig. 9 is here cited before Fig. 6 and following. Please re-order the figures accordingly.

- p.5 l.13, please correct '(Fig. X)' to Fig. 9 — or rather Fig. 6, see my previous comment.

- same line, the enrichment in carbon is very weak, and as the indeterminate tissue is apparently preserved as a relief, it could be as likely that the very weak contrast observed here rather reflects topography than a real chemical difference. Moreover, with the clear presence of calcium and phosphorous (i.e. very likely a mineralization into calcium phosphate), carbon is not really expected to be there.

- same line, phosphate should be phosphorous.

- p.5 l.14, ref. [61] (actually 59) is not the best references for primary phosphatization of tissues, and anyway, more argument than the presence of calcium and phosphorous are needed to justify that they constitute a primary mineral phase.
- p.5 l.20-21, the meaning of 'we added three species from their larval dataset' is unclear to me. Which dataset are you referring to here?
- Also, in the Supplementary Material text file (not identified as 'Supplementary Material 2' as p.5 l.23) the list of characters shows 213 characters, and not 214 as we would expect from 210 (p.5 l.19) + 4 (p.5 l.23)...
- Anyway, the final number of taxa and characters should appear clearly.
- p.5 l.23-24, binary is a multistate, please rephrase.
- p.5 l.41, von Siebold
- p.5 l.49, I understand why you used the suffix 'caris', but considering the position of Fibulocaris in your phylogeny (outside of Pancrustacea), associating Fibulocaris to crustaceans (which they are clearly not) could be highly misleading.
- p.6 l.3, please keep in the diagnosis only what is specific to the new taxon. 'Ovoid and slender bivalved carapace fused dorsally and frontally' applies to other taxa.
- p.6 l.8, don't you mean 'the carapace height/width greatly exceeds the total body length'? Or please rephrase.
- p.6 l.11-12, 'projecting the mouth and a pair of stalked eyes posteriorly' is not diagnosis.
- p.6 l.14, Fibulocaris nereidis sp. nov.
- p.6 l.19-21, Besides the holotype, shouldn't all figured specimens be paratypes?
- p.6 l.30, no decimal for maximum length and height whereas you used decimals before, so it should also be the case here. Are these maximum length and height from the same specimen? Please indicate specimen.s number.s?
- p.6 l.36, why don't you also cite Fig. 1 here?
- p.6 l.45, looking at the figures, especially the reconstruction Fig. 6, much less than 'a third of the thoracic segments and the telson' are uncovered. Especially as 'a clear differentiation between head and thorax cannot be inferred' (p.7 l.50).
- p.7 l.4-5, the 'presence of different morphotypes' is not discussed in the Taphonomy section as suggested by the authors.
- p.7 l.15, please legend these folds in Fig. 4A.
- p.7 l.18, I am quite sure that ref. [65] should be 64, and so from here (or maybe a bit before) there is only one number difference in references numbering.
- p.7 l.23, please explain/better describe what you mean by the 'differential position of the legs'
- p.7 l.41, you refer to Figure 1F, but there is no Figure 1F; Figure 1 stops at E.
- p.7 l.47, at this stage it is premature to say that Fibulocaris and Nereocaris are closely related as you didn't describe the results of your phylogenetic analyses.
- p.8 l.7, no Fig. 1F.
- p.8 l.26, citing only ref [43] is not enough here to refer to 'arthropods'.
- p.8 l.30, putative 'h.c.' are shown in Fig. 3F, not 2F.
- p.8 l.41, Figure 4B, C would need a close-up of the telson to illustrate the authors statement.
- p.8 l.44, you refer here to Figures 1A, B; 3B, C, but 3D actually shows the clearest view of the telson.
- p.8 l.46, the photograph in 4C is very dark, and the features are hard to see. A drawing is needed here.
- p.8 l.50, '3D subcircular traces'; moreover, are they fully 3D, i.e. spherical? Or do they just show some relief compared to the rest of the fossil?
- p.9 l.21, at least this Fibulocaris nereidis should be Fibulocaris nereidis gen. et sp. nov.
- p.9 l.31, it is unclear whether the 'frontal processes' refer to the long posteriorly-directed ventral spine, or to something else?
- p.9 l.44-45, 'thus leaving the Isoxyiidae as the only non-mandibulate bivalved group' please refer to figure 8. More generally you should refer more often to your figures even in the discussion.

- p.10 l.29, correct 'ventro-posterior spine'
- p.10 l.42-45, what about the 'indeterminate' tissue in Fig. 9B-D? Are nervous tissues in Burgess Shale fossils preserved in calcium phosphate -and so be the same here- or into something else?
- p.11 l.3, shouldn't ref. [83] be 82 here?
- p.11 l.21, I am quite sure that ref. [92] should be 83. But be careful, because you cite no reference between [83] p.11 l.3 and this [92], and you only have 90 references in your references list...
- p.11 l.45, here ref [84] is correct, and so are references from this point.
- p.11 l.53, please order the taxa following the respective reference numbers.
- p.11 l.56, ref. [87] doesn't deal with partially phosphatized 3D guts in arthropods.
- p.12 l.6, please add the reference corresponding to '+ref'.
- p.12 l.17, please add a reference for inverse notation in arthropods.
- p.12 l.18-19, please check whether ref. [86] is correct.
- p.12 l.23-24, please add a reference supporting inverted swimming in *Odaraia*. I guess [51]?
- p.12 l.33, neither 'are' all filter feeders.
- p.12 l.37, there is no ref. [93] in your references list.
- p.12 l.43, correct 'posterior-ventral spine'.
- p.12 l.44, do you really mean 'extant crustaceans' here, or rather 'extant arthropods'?
- p.12 l.49, 'we show a potential case', I much appreciate the prudence on the authors in their conclusions, but as said above I regret that this didn't extend into the title and abstract.
- p.13 l.6-7, 'a disparity THAT has started TO be unveiled'
- p.13 l.7, I guess that '=' should be references? Please add them.
- p.13 l.12-18, Please rewrite this section, which has clearly not been proofread: repetition of 'carapace' l.12, not elegant 'confirming this idea is still a work in progress' l.13, unclear 'as is a great understanding' l.16, ref. '[Daley]' not properly cited l.17 (and not present in the references list), misplaced coma between '[Daley]' and 'non-crustaceomorph' l.17. Described also more precisely what you mean by 'species recognized as having shields' because some taxa already cited such as the radiodonts can be considered as having a shield...

- please check carefully the references list, as I have noticed several errors such as:

ref. [10] should be Legg and co-authors.

ref. [26], title should be 'Crustacea Phyllopora and Malacostraca: a reappraisal of cephalic and thoracic shield and fold systems and their evolutionary significance'

ref. [40], for which Burgess Shale is spelled without capital letters.

ref. [71], 'Mittmann B& SG.' should be 'Mittmann B, Scholtz G.'

And I imagine several others...

- Fig. 1, legend mo(uth) doesn't show the same structure in A, B and E, could you please be more precise.

Also, scale bar in C is barely visible, could you perhaps use a white colour instead of black?

White scale bars should probably be used for each photograph.

- Fig. 2 caption, specimen shown in C is indicated as the counterpart of specimen in A, but they are clearly the same. C is just a close-up of A, isn't it?

Explain also, as you do in Fig. 3 caption, why the eye is visible in C but not A ('different polarization').

Also, a box locating the area shown in F is needed on photograph E.

- Fig. 3, scale bars in A, C, D, F and G should be white.

Also, line drawings of the telson would be useful as it appears very dark and the caudal rami (c.r.) is particularly hard to see in the photographs (see above).

More importantly, there is a clear labelling problem in valves orientation between photograph A and B, where the same valve is labelled l.v in A and r.v. in B. Considering that you have a ventral view (and I concur with this interpretation as one would expect to see the hinge in dorsal view, which is clearly not the case here, even though you needed to prepare out the spine on one side, which was very likely because dorso-ventral flattening of this specimen resulted in the spine to

be squished between the valves) then you should look at the part, and the valve appearing on top of photograph A is indeed the l.v, therefore so should also be the one on top of photograph B. Finally, boxes locating the close-ups shown in C-F are needed on photograph B, absolutely required for F.

- Fig. 4 caption, the use of 'specimens with 'highly 'preserved guts 'is unclear to me. Don't you actually mean 'specimens with 'the best 'preserved guts'?

- Fig. 5, scale bar in D should be white.

Also, a box locating the area shown in H is needed on photograph F.

- Fig. 5 caption, based on the apparent symmetry between specimens in E and G, it looks like the specimen shown in G. should be ROMIP 65379 'counterpart'.

Also, abbreviation le) is indicated as to be 'leg 'here, while you indicate it as 'legs 'in other captions, and you obviously show also several here.

Close-ups on the c.f. in I and s.f. in K are needed as they are otherwise absolutely not visible.

- Fig. 6 lacks letters accompanying the different diagrams, especially as the letters used in the caption do not follow the order of the diagrams in the figure. I would put the frontal view first (A), then lateral reconstruction (B), cross section through the cephalic region (c) and finally cross-section through the thorax (D). And don't forget to label the lines locating the cross-sections in the lateral view as C and D.

- Fig. 6 caption, A and B (or let's say the first to left diagrams in Fig. 6) are cross-sections across the cephalic and thoracic regions respectively. C (the third diagram from the left) is the frontal view. Please correct that.

- Fig. 7 caption, it may be worth indicating that you represented Fibulacaris during inverted swimming.

- Fig. 8, the use of a silhouette of Anomalocaris in front of the Isoxyiids is misleading.

Also, the blue dot with the silhouette of Fibulacaris appearing in front of Nereocaris briggsi is a bit misleading to the reader.

- Fig. 8 caption, see above but this tree is clearly not a 'simplified version 'of the tree shown in suppl. fig. but more an 'interpretative tree based on your consensus tree 'Please rephrase the title.

- Fig. 9 caption, D) caption shouldn't be in bold. And phosphate in D) and E) should be 'phosphorous 'as you refer here to the element, not the ion or compound.

Also, you say that the matrix comprises 214 characters, but there are only 213 in suppl- material (see above), but indeed 214 in the .xlsx and .nex files.

I hope that my comments will be useful, and help to further improve the manuscript.

Pierre Gueriau

Decision letter (RSOS-191350.R0)

08-Sep-2019

Dear Mr Izquierdo-López,

The editors assigned to your paper ("Inverted swimming in a new bivalved arthropod with a conspicuous ventro-posterior spine from the Cambrian Burgess Shale") have now received comments from reviewers. We would like you to revise your paper in accordance with the referee and Associate Editor suggestions which can be found below (not including confidential reports to the Editor). Please note this decision does not guarantee eventual acceptance.

Please submit a copy of your revised paper before 01-Oct-2019. Please note that the revision deadline will expire at 00.00am on this date. If we do not hear from you within this time then it will be assumed that the paper has been withdrawn. In exceptional circumstances, extensions may be possible if agreed with the Editorial Office in advance. We do not allow multiple rounds of revision so we urge you to make every effort to fully address all of the comments at this stage. If deemed necessary by the Editors, your manuscript will be sent back to one or more of the original reviewers for assessment. If the original reviewers are not available, we may invite new reviewers.

- Data accessibility

<http://datadryad.org/submit?journalID=RSOS&manu=RSOS-191350>

- Competing interests

- Authors' contributions

All submissions, other than those with a single author, must include an Authors' Contributions section which individually lists the specific contribution of each author. The list of Authors

should meet all of the following criteria; 1) substantial contributions to conception and design, or acquisition of data, or analysis and interpretation of data; 2) drafting the article or revising it critically for important intellectual content; and 3) final approval of the version to be published.

- Acknowledgements

- Funding statement

on behalf of Professor Rachel Wood (Associate Editor) and Kevin Padian (Subject Editor)
openscience@royalsociety.org

Editor comments:

We have two extensive and very useful reviews. The reviewers are overall positive but recommend a considerable number of changes. I encourage the authors to attend to the many points raised, and submit a revised manuscript. In particular please make sure to confine the Diagnosis to features that are unique to the taxon, not found in other taxa as well, as appropriate.

Comments to Author:

Reviewers' Comments to Author:

Reviewer: 1

Comments to the Author(s)

Title. The title should not contain the interpretation/speculation about its swimming posture up front and instead should reflect the fact that the bulk of this paper is a description of a new arthropod with a remarkable antero-ventral feature. The present versions comes across as attention-seeking.

Abstract. The first sentence can be deleted. Oblong in which view, plan view or side view? “Fused anteriorly” means what? The captions use legs, not limbs. “Bristle-like limbs” means what exactly? It is unlikely that a biramous arthropod leg would have the shape of a bristle. The abstract should not use we and our. It is not necessary to include the phylogenetic analysis since it does not show anything special. “Retrieves” and “recovers” may both be used in cladistics, but considering the average reader, they are awkward verbs for the suggested position in the tree.

Introduction. This is much too long and over-referenced. It should set the stage relatively briefly for the paper which is mostly descriptive, rather than being a long story about the nature and evolution of the carapace; some of this would better belong in the discussion, but better focused. Some carapaces are fused. Please avoid the all-inclusive “our”. The text cannot speak for everyone.

Taphonomy. Thin and “very thin” cannot be distinguished. This section mixes observations with interpretations. This is best avoided. Why does preservation with the animals lying on their sides indicate they “were normally turned sideways during transport”? What is the evidence for transport rather than the carcasses just lying or landing on their sides upon death? “Close bedding planes”? But the valves were fused were they not? The “ventral spine” seems like a long and recurved (or antero-ventrally curved) tapering rostrum – perhaps calling it a ventral spine is the wrong term. That it is not “normally showing breakage” indicates that sometimes it does – is this true, or should this be reworded? The “slight separation of the spine away from the ventral margin,” could this be just a morphological variation in the curvature rather than deformation? It seems the spine is stout and its base wide as it merges into the carapace, so breakage there might seem improbable. How straight is “very straight”? (In other words, please avoid using indefinable adjectives.) “Strikingly black” is another. The degree of upturning cannot be “strong”. What explanations other than moulting could there be for this attitude? So, what is the interpretation of the black rodlike feature? Does “carbon, calcium and phosphate elements” mean these as elements or something else? Calcium and phosphorus do not occur as elements in these shales.

The phylogenetic tree is not needed here. It does not differ all that much from Figure 3 of Legg et al. (2012), Figure 6 of Legg and Vannier (2013), and Vannier et al. (2018) in the relationships between the taxa of interest in the colored boxes, and the rest of the tree for the Pancrustacea etc. is not particularly relevant to the new form. The additional species incorporated in it do not add any real insight to the appreciation of Fibulacaris.

Systematic. Panarthropoda and Arthropoda should not be italicized. Shorten the genus etymology. The diagnosis for the genus is too long: it is more like a mini-description. In any case, since the genus is monospecific, here best is to put “as for the species, by reason of monotypy.” Line 14 need the genus epithet. Since there are 47 specimens, the complete list should be provided. In the opinion of this reviewer, it is not necessary to select two paratypes out of the whole collection. The diagnosis should stand alone (and be brief). This is because the species in the tangible object whereas the genus is a concept, not a thing. Line 24 gives a different number of specimens than in the Material section. Is there a reason why it is named after nereids?

Carapace. This could be a little shorter, and in places better worded. It is probably not necessary to refer to all the figures in detail if they are listed under the species name. This is unclear: “Both valves connect dorsally and extend slightly into a small dorsal process”. Where is this, considering the ventral portion of the carapace becomes the downward curving spine? The posteriorly projecting processes on the carapace are best not termed “ventral processes” but posterolateral processes or spines as one would with phyllocarids. The carapace hinge is fused so it is not quite right to uses valves in the plural. This is confusing: “anterior part is rotated ventro-

posteriorly into a single conspicuous spine" because a carapace does not rotate; later the abstract states the body is rotated, so this is confusing. Elsewhere the body is rotated. The dorsal crest or keel may be vaguely shown in Figure 4A but it does not seem obvious in the other images, and line 17 is vague about this. Maybe there is no crest and the reconstruction is wrong in this regard. An axial crest might seem somewhat unlikely, even if the valves are fused.

Structure – why structure and why not include this in the section above? "Reinforce the idea" should be reworded. "Cephalic fold" is also an unconventional term for grooves or furrows on the carapace. Carapace fold. Spine folds. The description of possible burial-induced wrinkles should use clearer terms and not imply that they are a carina or furrow. Under Head there is no mention of folds. Perhaps all the things labelled folds may not be folds, whatever is meant by the term. All this needs to be clarified. Interpretation is creeping into this section and it should be removed and put with the discussion.

Thorax. What is the evidence that *Nereocaris* is closely related, and how close is that? This is discussion and does not belong in the species description. Same with interpretation of the gut in the next paragraph, and referring to the caudal rami as weakly sclerotized.

The different subsections should be partly amalgamated.

Discussion. The valve-thorax shape is common in many taxa. "Could bear some resemblance" is awkward. So is "clear defensive function". "Rostral spine" appears here for the first time – it is a better term than ventral spine and "ventro-posterior spine" which comes later in this section. Again "highly elongated" and "highly multisegmented" are descriptions that have no agreed-upon scale; "closely related" is not appropriate. The carapace is not truly bivalved if it is fused, and presumably it was, in order for a rostrum to taper into a long curving spine. Some comparison with the trilobites that have a long spine projecting from the anterior border would be worthwhile. Also perhaps to trilobite hypostomes; some bear posteriorly directed projections.

Mode of Life. This section can be tightened. This reviewer is not certain that line 35 is correct, that the 3D nature of the gut is as critical. Gut contents to some extent (i.e. organic matter versus ingested sediment). How could this happen: "postmortem mud-infilled ingestions of sediment"? The upside-down swimming is worth suggesting but it is basically a suggestion or speculation, and "argue" (line 34) is not appropriate given the rationale in the paragraph. Carolinites is not much of an analog and the genal spines are posteriorly directed; the free cheek is angled ventrally. McCormick and Fortey did not make a case that it swam upside-down, so line 16 is just a speculation made in passing here, and does not support the case. Other possible interpretations apart from being inverted?

This discussion should be more concise and more logically organized. At present it jumps around. It needs to be careful in separating sensible interpretation from outright speculation, with a reduction of the latter.

Bibliography (should be References). The journal states it is not fussy about format and they will put the entries into their own house style, but this list makes little attempt to follow it and comes across as somewhat sloppy. What does "internet" refer to? In any case, there are too many references for a paper like this. The number can be reduced by maybe a third. For example, in the Introduction basic statements are supported by up to half a dozen citations.

Figure captions. It seems that a bolded title is not needed. "Highly preserved" should be well preserved. Close-up of, not on. The captions should be more explanatory, not just a list of specimen numbers. This would help the reader immeasurably.

Images. These are good. They should all be in black and white to get the tones uniform.

Figure 1. Probably just B is needed as the dry version A does not show any additional feature or the structures more clearly. C is dry? Is C needed? Which portion of the gut in A/B? Since these are all from the same specimen it is not necessary to repeat the number.

Figure 2. It might be an idea to flip B and D so that it can be compared with A. In B, how certain is the right eye? It could be labelled in C. Is D dry? Gut displacement could have a different name, like flexure. Valves?

Figure 3. What is the black blob at the anterior end? Possible is better than putative. The three features in F do not look like cavities. Setae? The black objects might be eggs but the text refers to them as "traces". D and E are not both wet? The location of F should be noted (with an arrow). Is it necessary to label a feature "indeterminate"?

Figure 4. Lateral views. B is superfluous, and maybe C is not useful. "Part and counterpart overlap" means what? Which part is shown in H?

Figure 6. The letters are too small. Need C and D. The photographic images did not specifically show the dorsal ridge or crest. The caption is too brief. Maybe a ventral view too.

Figure 7 should have been completed before submission.

Figure 8. Fibulacaris is considered closely related to Nereocaris but on the basis of what evidence? The caption is too brief. It should also explain the colors etc. What group is Malacrustacea? This is not a taxon in common usage.

Figure 9 should just show Fibulacaris. Page 12 refers to Marrella but this is not relevant and the comparison vague.

Reviewer: 2

Comments to the Author(s)

Manuscript RSOS-191350 reports on a new bivalved arthropod from the renewed Cambrian Burgess Shale, which is clearly different from any other fossils described so far and as such definitely deserves to be published. The fossil description is neat and rather well illustrated, and the text is overall well organized and well written, yet the manuscript (particularly the references and the conclusion) is unfinished, and there are (in my opinion) some issues, listed below, that need to be addressed before the manuscript can be considered for publication.

Major comments:

(Note that I refer to page and line numbers provided in the pdf I had access to online, which do not correspond to real line numbers.)

- While the authors are very careful in their conclusions in the main text, making common use of 'potential.ly', 'possible.bly', etc. their title and abstract are not, and I would suggest them to keep careful therein as well. I am here particularly referring to the title and the use of 'confirms' in the last sentence of the abstract, while the authors are way more careful in their discussion.

- In the title, as well as in many places along the text, the authors refer to the long ventral spine of their fossil as a 'ventro-posterior spine' (even 'posterior-ventral spine' p.12 l.43), but this spine is NOT ventro-posterior because it is actually positioned at the anterior end of the animal but is directed towards the posterior. I would rather use 'posteriorly-directed ventral spine'.

- There is a big mess with references numbering (a bit detailed below), which absolutely needs to be carefully checked before publication.

- As the fossils are quite small, the authors state that some anatomical features such as the limbs are poorly preserved or at least hard to observe. This is even more the case in some of the photographs, and I deplore the lack of drawings to help the reader identifying some of the anatomical features described. Similarly, (except for fig. 9) close-ups are not associated with boxes in the corresponding large-scale photographs and as such are often hard to locate...

- Authors for the taxa are not given. Please add them when you first mention a taxon.

- The phylogeny is probably the main issue of this paper. Besides the fact that Aria & Caron's phylogenies (on which this work is based) are far from making consensus among the community, I noticed the following issues with the way the analysis was performed and the data are presented:

(1) The authors only present a simplified version of a consensus tree from a Bayesian analysis (plus one tree as suppl. fig.), but totally discard parsimony analyses, which in my opinion should be also produced at least for comparison.

(2) The 'simplified version of a consensus tree' is actually NOT a simplified version, as it shows relationships different than in the complete tree presented as suppl. figure; for instance, radiodonts appear monophyletic in Fig. 8, but are absolutely not in the complete tree is suppl. Fig., which is another serious issue...

(3) Each new paper using Aria & Caron's phylogeny show important topological changes among the trees, questioning its stability. Here, there is also notable differences with the tree published by Vannier et al. 2018, on which the present analysis is largely based (in particular the position of Hymenocarina and Myriapoda). This should be discussed much more than the couple of sentences p.9 l.41-48. And the new topology should also be compared to those in both papers published by Aria & Caron in 2017 (your refs [4, 5]).

(4) The 'four new characters' added (p.5 l.23) should be described in the main text, and the effect of their addition at least discussed (showing trees with and without these characters, at least in Suppl. Material, would be very useful). Note that they are not much discussed in Suppl. Material either.

(5) The backbone constraint used (p.5 l-3) can only be found in the matrix nexus file, but nowhere else in the main text or suppl. material.

- Fig. 3A, the long posteriorly-directed ventral spine seems segmented/annulated, is this due to the quality of the photograph in the pdf version, or could it be a real feature?

- And a totally open question: Considering that Fibulacaris specimens are quite small (maximum length: 2 cm) compared to most other bivalved arthropods from the Burgess Shale, and considering also that conspicuous spines are common in larval/juvenile crustacean stages, couldn't Fibulacaris represent a larval/juvenile form of a larger bivalved arthropod found in the same layers? Of course, the putative eggs (an unlikely interpretation in my opinion) would make them mature individuals, but this constitutes a weak (and very unlikely) argument to rule out the possibility that they were larval/juvenile forms.

Additional comments / corrections:

(Note that I refer to page and line numbers provided in the pdf I had access to online, which do not correspond to real line numbers.)

- p.2 l.27, the convergence with trilobites that appears here in the abstract is only superficially discussed (in a single sentence) in the text, so the comparison appears here very weak.

- p.2 l.33, the interesting conclusion/implication that the bivalved carapace may have been a basal trait for Mandibulata could be included into the abstract.

- p.3 l.7, please illustrate the different type of carapace covering by giving names of modern organisms showing such features.

- p.3 l.14, please indicate a date for the Anostraca, and rather refer to a fossil than to a molecular clock that we know is hard to trust... Actually, the Cambrian *Rehbachella* (stem-group Branchiopoda) had a carapace, and the oldest-known fossil anostracans (*Lepidocaris rhyniensis* and *Haltinnai serrata*) are from the Devonian (Early and Late Devonian, respectively).

- p.3 l.36, here starts the mess with references numbering. I am quite sure that ref. [50] should be 48.

- p.3 l.37, I am quite sure that ref. [51] should be 49.

- p.3 l.39, I am quite sure that ref. [52] should be 50.

Check all the references numbering from this point, as the problem persists almost all along the rest of the text.

- p.3 l.39, 'as well AS feeding modes'

- p.4 l.16-17, '(ROMIP) Collections'.

- p.4 l.53, Fig. 9 is here cited before Fig. 6 and following. Please re-order the figures accordingly.

- p.5 l.13, please correct '(Fig. X)' to Fig. 9 – or rather Fig. 6, see my previous comment.

- same line, the enrichment in carbon is very weak, and as the indeterminate tissue is apparently preserved as a relief, it could be as likely that the very weak contrast observed here rather reflects topography than a real chemical difference. Moreover, with the clear presence of calcium and phosphorous (i.e. very likely a mineralization into calcium phosphate), carbon is not really expected to be there.

- same line, phosphate should be phosphorous.

- p.5 l.14, ref. [61] (actually 59) is not the best references for primary phosphatization of tissues, and anyway, more argument than the presence of calcium and phosphorous are needed to justify that they constitute a primary mineral phase.

- p.5 l.20-21, the meaning of 'we added three species from their larval dataset' is unclear to me. Which dataset are you referring to here?

Also, in the Supplementary Material text file (not identified as 'Supplementary Material 2' as p.5 l.23) the list of characters shows 213 characters, and not 214 as we would expect from 210 (p.5 l.19) + 4 (p.5 l.23)...

Anyway, the final number of taxa and characters should appear clearly.

- p.5 l.23-24, binary is a multistate, please rephrase.

- p.5 l.41, von Siebold

- p.5 l.49, I understand why you used the suffix 'caris', but considering the position of *Fibulocaris* in your phylogeny (outside of Pancrustacea), associating *Fibulocaris* to crustaceans (which they are clearly not) could be highly misleading.

- p.6 l.3, please keep in the diagnosis only what is specific to the new taxon. 'Ovoid and slender bivalved carapace fused dorsally and frontally' applies to other taxa.

- p.6 l.8, don't you mean 'the carapace height/width greatly exceeds the total body length'? Or please rephrase.

- p.6 l.11-12, 'projecting the mouth and a pair of stalked eyes posteriorly' is not diagnosis.

- p.6 l.14, *Fibulocaris nereidis* sp. nov.

- p.6 l.19-21, Besides the holotype, shouldn't all figured specimens be paratypes?

- p.6 l.30, no decimal for maximum length and height whereas you used decimals before, so it should also be the case here. Are these maximum length and height from the same specimen? Please indicate specimen.s number.s?
- p.6 l.36, why don't you also cite Fig. 1 here?
- p.6 l.45, looking at the figures, especially the reconstruction Fig. 6, much less than 'a third of the thoracic segments and the telson' are uncovered. Especially as 'a clear differentiation between head and thorax cannot be inferred' (p.7 l.50).
- p.7 l.4-5, the 'presence of different morphotypes' is not discussed in the Taphonomy section as suggested by the authors.
- p.7 l.15, please legend these folds in Fig. 4A.
- p.7 l.18, I am quite sure that ref. [65] should be 64, and so from here (or maybe a bit before) there is only one number difference in references numbering.
- p.7 l.23, please explain/better describe what you mean by the 'differential position of the legs'
- p.7 l.41, you refer to Figure 1F, but there is no Figure 1F; Figure 1 stops at E.
- p.7 l.47, at this stage it is premature to say that Fibulacaris and Nereocaris are closely related as you didn't describe the results of your phylogenetic analyses.
- p.8 l.7, no Fig. 1F.
- p.8 l.26, citing only ref [43] is not enough here to refer to 'arthropods'.
- p.8 l.30, putative 'h.c.' are shown in Fig. 3F, not 2F.
- p.8 l.41, Figure 4B, C would need a close-up of the telson to illustrate the authors statement.
- p.8 l.44, you refer here to Figures 1A, B; 3B, C, but 3D actually shows the clearest view of the telson.
- p.8 l.46, the photograph in 4C is very dark, and the features are hard to see. A drawing is needed here.
- p.8 l.50, '3D subcircular traces'; moreover, are they fully 3D, i.e. spherical? Of do they just show some relief compared to the rest of the fossil?
- p.9 l.21, at least this Fibulacaris nereidis should be Fibulacaris nereidis gen. et sp. nov .
- p.9 l.31, it is unclear whether the 'frontal processes' refer to the long posteriorly-directed ventral spine, or to something else?
- p.9 l.44-45, 'thus leaving the Isoxyiidae as the only non-mandibulate bivalved group' please refer to figure 8. More generally you should refer more often to your figures even in the discussion.
- p.10 l.29, correct 'ventro-posterior spine'
- p.10 l.42-45, what about the 'indeterminate' tissue in Fig. 9B-D? Are nervous tissues in Burgess Shale fossils preserved in calcium phosphate -and so be the same here- or into something else?
- p.11 l.3, shouldn't ref. [83] be 82 here?
- p.11 l.21, I am quite sure that ref. [92] should be 83. But be careful, because you cite no reference between [83] p.11 l.3 and this [92], and you only have 90 references in your references list...
- p.11 l.45, here ref [84] is correct, and so are references from this point.
- p.11 l.53, please order the taxa following the respective reference numbers.
- p.11 l.56, ref. [87] doesn't deal with partially phosphatized 3D guts in arthropods.
- p.12 l.6, please add the reference corresponding to '+ref'.
- p.12 l.17, please add a reference for inverse notation in arthropods.
- p.12 l.18-19, please check whether ref. [86] is correct.
- p.12 l.23-24, please add a reference supporting inverted swimming in Odaraia. I guess [51]?
- p.12 l.33, neither 'are' 'all filter feeders.
- p.12 l.37, there is no ref. [93] in your references list.
- p.12 l.43, correct 'posterior-ventral spine'.
- p.12 l.44, do you really mean 'extant crustaceans' here, or rather 'extant arthropods'?
- p.12 l.49, 'we show a potential case', I much appreciate the prudence on the authors in their conclusions, but as said above I regret that this didn't extend into the title and abstract.
- p.13 l.6-7, 'a disparity THAT has started TO be unveiled'
- p.13 l.7, I guess that '=' should be references? Please add them.

- p.13 l.12-18, Please rewrite this section, which has clearly not been proofread: repetition of 'carapace' l.12, not elegant 'confirming this idea is still a work in progress' l.13, unclear 'as is a great understanding' l.16, ref. '[Daley]' not properly cited l.17 (and not present in the references list), misplaced comma between '[Daley]' and 'non-crustaceomorph' l.17. Described also more precisely what you mean by 'species recognized as having shields' because some taxa already cited such as the radiodonts can be considered as having a shield...

- please check carefully the references list, as I have noticed several errors such as:

ref. [10] should be Legg and co-authors.

ref. [26], title should be 'Crustacea Phyllopora and Malacostraca: a reappraisal of cephalic and thoracic shield and fold systems and their evolutionary significance'

ref. [40], for which Burgess Shale is spelled without capital letters.

ref. [71], 'Mittmann B& SG.' should be 'Mittmann B, Scholtz G.'

And I imagine several others...

- Fig. 1, legend mo(uth) doesn't show the same structure in A, B and E, could you please be more precise.

Also, scale bar in C is barely visible, could you perhaps use a white colour instead of black?

White scale bars should probably be used for each photograph.

- Fig. 2 caption, specimen shown in C is indicated as the counterpart of specimen in A, but they are clearly the same. C is just a close-up of A, isn't it?

Explain also, as you do in Fig. 3 caption, why the eye is visible in C but not A ('different polarization').

Also, a box locating the area shown in F is needed on photograph E.

- Fig. 3, scale bars in A, C, D, F and G should be white.

Also, line drawings of the telson would be useful as it appears very dark and the caudal rami (c.r.) is particularly hard to see in the photographs (see above).

More importantly, there is a clear labelling problem in valves orientation between photograph A and B, where the same valve is labelled l.v in A and r.v. in B. Considering that you have a ventral view (and I concur with this interpretation as one would expect to see the hinge in dorsal view, which is clearly not the case here, even though you needed to prepare out the spine on one side, which was very likely because dorso-ventral flattening of this specimen resulted in the spine to be squished between the valves) then you should look at the part, and the valve appearing on top of photograph A is indeed the l.v, therefore so should also be the one on top of photograph B. Finally, boxes locating the close-ups shown in C-F are needed on photograph B, absolutely required for F.

- Fig. 4 caption, the use of 'specimens with 'highly 'preserved guts 'is unclear to me. Don't you actually mean 'specimens with 'the best 'preserved guts'?

- Fig. 5, scale bar in D should be white.

Also, a box locating the area shown in H is needed on photograph F.

- Fig. 5 caption, based on the apparent symmetry between specimens in E and G, it looks like the specimen shown in G. should be ROMIP 65379 'counterpart'.

Also, abbreviation le) is indicated as to be 'leg 'here, while you indicate it as 'legs 'in other captions, and you obviously show also several here.

Close-ups on the c.f. in I and s.f. in K are needed as they are otherwise absolutely not visible.

- Fig. 6 lacks letters accompanying the different diagrams, especially as the letters used in the caption do not follow the order of the diagrams in the figure. I would put the frontal view first (A), then lateral reconstruction (B), cross section through the cephalic region (c) and finally cross-section through the thorax (D). And don't forget to label the lines locating the cross-sections in the lateral view as C and D.

- Fig. 6 caption, A and B (or let's say the first to left diagrams in Fig. 6) are cross-sections across the cephalic and thoracic regions respectively. C (the third diagram from the left) is the frontal view. Please correct that.

- Fig. 7 caption, it may be worth indicating that you represented Fibulacaris during inverted swimming.
- Fig. 8, the use of a silhouette of Anomalocaris in front of the Isoxyiids is misleading. Also, the blue dot with the silhouette of Fibulacaris appearing in front of Nereocaris briggsi is a bit misleading to the reader.
- Fig. 8 caption, see above but this tree is clearly not a 'simplified version' of the tree shown in suppl. fig. but more an 'interpretative tree based on your consensus tree' Please rephrase the title.
- Fig. 9 caption, D) caption shouldn't be in bold. And phosphate in D) and E) should be 'phosphorous' as you refer here to the element, not the ion or compound. Also, you say that the matrix comprises 214 characters, but there are only 213 in suppl- material (see above), but indeed 214 in the .xlsx and .nex files.

I hope that my comments will be useful, and help to further improve the manuscript.

Pierre Gueriau

Author's Response to Decision Letter for (RSOS-191350.R0)

See Appendix A.

Decision letter (RSOS-191350.R1)

22-Oct-2019

Dear Mr Izquierdo-López,

I am pleased to inform you that your manuscript entitled "A possible case of inverted lifestyle in a new bivalved arthropod from the Burgess Shale" is now accepted for publication in Royal Society Open Science.

Kind regards,
Lianne Parkhouse
Editorial Coordinator
Royal Society Open Science
openscience@royalsociety.org

on behalf of Professor Rachel Wood (Associate Editor) and Kevin Padian (Subject Editor)
openscience@royalsociety.org

Appendix A

We would like to thank the time and dedication that both reviewers have spent on this manuscript, and we ascertain that their comments have greatly improved our manuscript. We would like to address each of the reviewer's comments individually, as follows:

Reviewer 1:

Title. The title should not contain the interpretation/speculation about its swimming posture up front and instead should reflect the fact that the bulk of this paper is a description of a new arthropod with a remarkable antero-ventral feature. The present versions comes across as attention-seeking.

We have redrafted the title to acknowledge that ecological reconstructions for extinct taxa are, in nature, speculative, including invert swimming. We believe that this type of title can attract readers from many different disciplines, a target of Royal Society Open, rather than using a descriptive title, more suited for arthropod workers.

Abstract.

The first sentence can be deleted.

We consider the first sentence of the abstract to give a neat introduction to arthropod carapace and bivalved arthropods for readers not familiar with this feature, but overall, we have revised it and followed both reviewer's suggestions and reshaped the whole abstract.

Oblong in which view, plan view or side view? "Fused anteriorly" means what?

Oblong in side view, fused anteriorly and dorsally as in that that both valves are fused on the anterior part of the animal, so that the head is frontally covered by the carapace.

The captions use legs, not limbs. "Bristle-like limbs" means what exactly? It is unlikely that a biramous arthropod leg would have the shape of a bristle.

We have corrected limbs to legs.

We have deleted "bristle-like limbs".

The abstract should not use we and our.

We have eliminated the use of "our" and "we" all throughout the text, as suggested.

It is not necessary to include the phylogenetic analysis since it does not show anything special.

Please refer to our comments on the phylogenetic analysis later.

"Retrieves" and "recovers" may both be used in cladistics, but considering the average reader, they are awkward verbs for the suggested position in the tree.

We have changed the term "retrieves", as suggested

Introduction.

This is much too long and over-referenced. It should set the stage relatively briefly for the paper which is mostly descriptive, rather than being a long story about the nature and evolution of the carapace; some of this would better belong in the discussion, but better focused.

We have taken a critical look and managed to reduce this section. We still wanted to convey an introduction to the origin and morphological disparity of the carapace. For this reason, we wanted to have a bigger introduction than just a taxonomic-orientated paper. We also think the average reader may find useful information to introduce themselves into the topic of Cambrian bivalved arthropods.

Some carapaces are fused.

As the reviewer suggested, some carapaces are fused. The nature and even the definition of a carapace, are, from our understanding complicated and we would like to restrict the idea of a carapace to Cambrian arthropods and crustaceans with analogous structures (see also Olesen, 2013) for our definition of a carapace, and we have added this notion in the text. Structures such as the prosoma of a chelicerate, or the cephalic shield are here not regarded as carapaces.

Please avoid the all-inclusive “our”. The text cannot speak for everyone.

We have eliminated the use of “our” and “we” all throughout the text, as suggested.

Taphonomy.

Thin and “very thin” cannot be distinguished.

Changed to “thin”, as suggested.

This section mixes observations with interpretations. This is best avoided.

We have revised this section.

Why does preservation with the animals lying on their sides indicate they “were normally turned sideways during transport”? What is the evidence for transport rather than the carcasses just lying or landing on their sides upon death?

Most bivalved arthropods from the Burgess Shale are found preserved in multiple views (eg. dorsal, ventral...), but in the case of *F. nereidis*, non-lateral views were extremely rare, which may suggest that the morphology of the carapace (flattened sideways, as seen in Figure 3) may be responsible for this phenomenon. We have deleted the word “transport” for clarification.

“Close bedding planes”? But the valves were fused were they not?

We were talking about the ventral and posterior margins of the valves which are not fused. We have rephrased as follow: ...” Left and right side of the carapace were narrowly separated along their ventral and posterior margins, as evidenced by specimens which preserve both sides of the carapace on top of each other (e.g. Figures 2E; 4A; 5I) and by specimens preserved in ventral or dorsal views (Figure 1C, D; Figure 3).”

The “ventral spine” seems like a long and recurved (or antero-ventrally curved) tapering rostrum—perhaps calling it a ventral spine is the wrong term.

We agree with the interpretation of the reviewer on the rostrum. We have chosen to use the term “postero-ventrally recurved rostrum”.

That it is not “normally showing breakage” indicates that sometimes it does—is this true, or should this be reworded?

We have no evidence of breakage per se and we have clarified this point in our ms. We have a few specimens where the terminal end of the spine is not visible: this could be because the spine is

indeed broken, but in such cases the tip seems to continue within the sediment and it is impossible to mechanically prepare it.

The “slight separation of the spine away from the ventral margin,” could this be just a morphological variation in the curvature rather than deformation? It seems the spine is stout and its base wide as it merges into the carapace, so breakage there might seem improbable.

In two extreme cases (e.g. Fig 4A, Fig 5D), a bump along the anterior of the carapace and just above the spine might suggest a soft deformation of the carapace, perhaps as a result of the spine being pulled away from its original position (in these two cases the rostrum seems rotated clockwise by 20-30 degrees, but the vast majority of all the other 100 specimens that we have shown the rostrum parallel to the ventral side of the carapace). We have added a note to the text as also suggested by Reviewer 2, acknowledging the possibility of population variations, although we strongly think that a taphonomic explanation is more likely and these two specimens represent outliers.

How straight is “very straight”? (In other words, please avoid using indefinable adjectives.) “Strikingly black” is another. The degree of upturning cannot be “strong”.

We have changed “very straight” to “straight”.

We have changed “strikingly black” to “black”

We deleted “strongly”

What explanations other than moulting could there be for this attitude?

We think these specimens probably represent moults, given that we lack features that are commonly preserved in other specimens, such as eyes and gut. We remain slightly cautious, though, as these could still represent carcasses in different stages of decay that lack these features due to differential taphonomy conditions, including burial angles.

So, what is the interpretation of the black rodlike feature? Does “carbon, calcium and phosphate elements” mean these as elements or something else? Calcium and phosphorus do not occur as elements in these shales.

The rod-like structures are probably the result of an initial phase of phosphatization of the gut contents, as commented in the discussion and postulated by other workers (Butterfield et al. 2007). We have rephrased this last part of the paragraph.

Systematic

The phylogenetic tree is not needed here. It does not differ all that much from Figure 3 of Legg et al. (2012), Figure 6 of Legg and Vannier (2013), and Vannier et al. (2018) in the relationships between the taxa of interest in the colored boxes, and the rest of the tree for the Pancrustacea etc. is not particularly relevant to the new form. The additional species incorporated in it do not add any real insight to the appreciation of *Fibulacaris*.

We agree with the reviewer that the phylogenetic tree does not represent the main focus of this article and our results do not differ strikingly from Vannier et al. (2018) in particular, although we argue it complements it (see also comments to reviewer 2). As it is customary for any publication of a new species, we are interested to put *Fibulacaris* into an evolutionary context and compare it with other Cambrian species. We think that the most objective way to achieve that is through a phylogenetic analysis and we think that the tree provides some interesting information on its

affinities that can be of high interest to arthropod workers, although of course acknowledging that it remains a hypothesis of relationship, which will need to be tested with further data in the future.

Panarthropoda and Arthropoda should not be italicized.

We have deleted the italics, as recommended.

Shorten the genus etymology. The diagnosis for the genus is too long: it is more like a mini-description. In any case, since the genus is monospecific, here best is to put “as for the species, by reason of monotypy.”

The genus etymology was shortened, we reduced the diagnosis and we have added “as for the species, by reason of monotypy.”

Line 14 need the genus epithet.

Added the genus epithet

Since there are 47 specimens, the complete list should be provided.

We have listed the catalogue numbers of all the material used in this study including for measurements in supplementary material.

In the opinion of this reviewer, it is not necessary to select two paratypes out of the whole collection.

Reviewer 2 suggested, instead, that all figured specimens should constitute paratypes. As far as we know there are no set rules on number of paratypes. We have chosen to retain only 2 paratypes, which we believe provide the most information.

The diagnosis should stand alone (and be brief). This is because the species in the tangible object whereas the genus is a concept, not a thing.

The diagnosis was reduced. We agree with the reviewer, but added the diagnosis to the genus

Line 24 gives a different number of specimens than in the Material section.

After careful check of our entire collections, the number of specimens of this new species has been revised to 102. However, not all specimens were used for the analysis (measurements and interpretation of the morphology). The other specimens were too weathered or poorly preserved to be useful for any of the analyses. This being said we have listed all specimens known of this species in supplementary information.

Is there a reason why it is named after nereids?

The closely related *Nereocaris*, was named after Nereus, father of the nereids, hence the epithet *nereidis* for *Fibulacaris*.

Carapace. This could be a little shorter, and in places better worded. It is probably not necessary to refer to all the figures in detail if they are listed under the species name.

We have revised the description of the carapace and clarified some statements. We would like to refer to our figures in the description, so that the reader can identify the features while associating the text to them, although we have reduced the number of references to them for clarity (i.e. using “e.g.”).

This is unclear: “Both valves connect dorsally and extend slightly into a small dorsal process”. Where is this, considering the ventral portion of the carapace becomes the downward curving spine?

This refers to the small rounded process that appears in the posterior part of the carapace, dorsally, labelled “d.p” in the figures. We have rephrased as follow: “The posterior dorsal margin of the carapace terminates into a small process.”

The posteriorly projecting processes on the carapace are best not termed “ventral processes” but posterolateral processes or spines as one would with phyllocarids.

We have changed the terminology as suggested.

The carapace hinge is fused so it is not quite right to uses valves in the plural

We have rephrased using the terms left and right sides of the carapace

This is confusing: “anterior part is rotated ventro-posteriorly into a single conspicuous spine” because a carapace does not rotate; later the abstract states the body is rotated, so this is confusing. Elsewhere the body is rotated.

The body itself is also recurved backward, we have clarified this point in the abstract and in the ms.

The dorsal crest or keel may be vaguely shown in Figure 4A but it does not seem obvious in the other images, and line 17 is vague about this. Maybe there is no crest and the reconstruction is wrong in this regard. An axial crest might seem somewhat unlikely, even if the valves are fused.

As is common in Burgess Shale material, specimens show a gradient of preservation for all features. We have used cautionary language (e.g. “may suggest”), but we support the presence of this feature also by comparison with similar features preserved in *Nereocaris* from the Burgess Shale, for example.

Structure—why structure and why not include this in the section above? “Reinforce the idea” should be reworded. “Cephalic fold” is also an unconventional term for grooves or furrows on the carapace. Carapace fold. Spine folds. The description of possible burial-induced wrinkles should use clearer terms and not imply that they are a carina or furrow.

We have deleted the subsection, as suggested

We have reworded the sentence

We follow the recommendation of the reviewer and changed, in the figures, cephalic folds for “crest” or “furrow” accordingly.

Under Head there is no mention of folds. Perhaps all the things labelled folds may not be folds, whatever is meant by the term. All this needs to be clarified. Interpretation is creeping into this section and it should be removed and put with the discussion.

As folds we mean burial-induced compressions of the carapace that may have created a slight overlap between parts of the carapace. These overlaps would be more common in parts of the carapace that naturally would show some changes in slope as is the case for things like a carina or furrow. The head, therefore does not have any fold (as for head, we only describe non-carapace structures) and we have tried to clarify the meaning of the folds in the ms. We have similarly addressed that in the figures.

Thorax.

What is the evidence that *Nereocaris* is closely related, and how close is that?

We acknowledge that the comparison is premature and have deleted it.

This is discussion and does not belong in the species description. Same with interpretation of the gut in the next paragraph, and referring to the caudal rami as weakly sclerotized.

We have addressed this point to the best of our ability.

We have changed “gut” to “structure inside the gut’s lumen”. The caudal rami being weakly sclerotized compared to the rest of the body (legs...) which are normally easier preserved in these specimens, can also be regarded as an observation, although this section was rewritten.

The different subsections should be partly amalgamated.

If referred to different descriptive subsections, we followed the same structure as in other articles from the same working group (for example *Waptia*, Vannier et al. 2018), separating the description into subsections that we think are useful to the reader in order to find information more rapidly.

Discussion.

The valve–thorax shape is common in many taxa.

We agree that the valve-thorax shape is common in many taxa, but we wanted to extend the similarities just to species present in our phylogeny, if not a rare trait.

“Could bear some resemblance” is awkward. So is “clear defensive function”. “Rostral spine” appears here for the first time—it is a better term than ventral spine and “ventro-posterior spine” which comes later in this section.

We changed “could bear some resemblance”.

We deleted the term “clear” and we replaced by “antipredatory function”.

We have replaced it to rostrum

Again “highly elongated” and “highly multisegmented” are descriptions that have no agreed-upon scale; “closely related” is not appropriate.

We have deleted the adverb “highly” in most cases and added comparison for “highly multisegmented”. For example, although *Plenocaris plena* is of a similar size to *Nereocaris exilis*, the first species presents ca. 15 thoracic-abdominal segments, while the latter has more than 30.

We do not understand why “closely related” should not be appropriate in this case, as *N.exilis* is sister group to *Fibulacaris* +(*Perspicares*+*Clypeocaris*).

The carapace is not truly bivalved if it is fused, and presumably it was, in order for a rostrum to taper into a long curving spine.

We refer as “bivalved” as a carapace that extends ventrally into two lateral sides, despite not having a true hinge. This nomenclature has also been used in similar cases (e.g. *Waptia*), although we acknowledge the problems this term can arise and therefore have written “bivalved” throughout the text.

Some comparison with the trilobites that have a long spine projecting from the anterior border would be worthwhile. Also perhaps to trilobite hypostomes; some bear posteriorly directed projections.

We have added some insights into the comparison with trilobites, as suggested

Mode of Life. This section can be tightened.

The section was reworked and has been tightened.

This reviewer is not certain that line 35 is correct, that the 3D nature of the gut is as critical. Gut contents to some extent (i.e. organic matter versus ingested sediment).

Correctly interpreting the 3D nature of the gut is one important aspect of their ecology, but we have acknowledged that it is only part of the story.

How could this happen: “postmortem mud-infilled ingestions of sediment”?

We refer to sediment ingestion while the animal was in turbidity flows, as suggested for other Burgess Shale arthropods (ie. *Naraoia*, in Vannier, 2002). We have changed the sentence for clarification.

The upside-down swimming is worth suggesting but it is basically a suggestion or speculation, and “argue” (line 34) is not appropriate given the rationale in the paragraph.

We have been more cautious in our suggestion and have used more cautionary language (changed “argue” to “may have been”).

Carolinites is not much of an analog and the genal spines are posteriorly directed; the free cheek is angled ventrally. McCormick and Fortey did not make a case that it swam upside-down, so line 16 is just a speculation made in passing here, and does not support the case.

We agree that McCormick & Fortey did not make a case that it swam upside-down, and therefore the reference appears before that statement. We have expanded on this section, nevertheless.

Other possible interpretations apart from being inverted?

The ventral position of the posteriorly-directed ventral spine could have made walking difficult, if the spine was not partially or totally buried in the sediment. The animal could have also been swimming not inverted, and perhaps living above the substrate on flocculent mud layers to be able to capture food particles, although there are more reasons to believe the former.

This discussion should be more concise and more logically organized. At present it jumps around. It needs to be careful in separating sensible interpretation from outright speculation, with a reduction of the latter.

We have reduced and improved the organization of the discussion.

We have used caution in all the ecological interpretations of the animal, while at the same time trying to convey a more interesting approach to uncover its lifestyle than just limiting to suspension feeding. *Fibulacaris* presents striking traits (such as the rostrum, or the anteriorly rotated head) that we regard of interest for any researcher interested in functional morphology and therefore have expressed our thoughts on the subject. Researchers, especially those studying extant species are welcome to challenge, improve or disregard our interpretations and advise us regarding which interpretations would need further support.

Bibliography (should be References). The journal states it is not fussy about format and they will put the entries into their own house style, but this list makes little attempt to follow it and comes across as somewhat sloppy. What does “internet” refer to?

Changed to “References”

We thank the reviewer for spotting this major issue after a problem with the citation program we used and we have now reworked on all our references.

In any case, there are too many references for a paper like this. The number can be reduced by maybe a third. For example, in the Introduction basic statements are supported by up to half a dozen citations.

We have reduced the introduction, and hence, the number of references.

Figure captions. It seems that a bolded title is not needed. “Highly preserved” should be well preserved. Close-up of, not on. The captions should be more explanatory, not just a list of specimen numbers. This would help the reader immeasurably.

The bolded title was deleted

Changed the sentence “highly preserved” to best-preserved

Changed to “Close-up of”

We want the reader to read the full description and associate it with the figures and therefore the captions only contain essential information such as acronyms and type of view (lateral...) although we have made an attempt to add more explanation of key features when relevant.

Images. These are good. They should all be in black and white to get the tones uniform.

We appreciate the suggestion of the reviewer, but we believe that changing the images to black and white will not improve their quality. Thankfully, the colours of the Marble Canyon shale are different grey tonalities that already allow for clear differentiation of the morphological features of the fossils.

Figure 1. Probably just B is needed as the dry version A does not show any additional feature or the structures more clearly. C is dry? Is C needed? Which portion of the gut in A/B? Since these are all from the same specimen it is not necessary to repeat the number.

We kept Figure 1A as it is and changed Figure 1B for a drawing

C is dry and we removed it.

Please note that the figure C and D have been replaced for ventral and dorsal views of two different specimens which we think complements nicely the holotype specimen figured in A and B.

Figure 2. It might be an idea to flip B and D so that it can be compared with A. In B, how certain is the right eye? It could be labelled in C. Is D dry? Gut displacement could have a different name, like flexure. Valves?

We applied the suggestions of the reviewer to flip B.

In figure B, the eye on the right is difficult to infer, but the counterpart (A) clearly shows a spherical structure on the spine that may correspond to this eye.

Figure C has been replaced.

Figure D has been replaced to reduce redundancy.

We acknowledge that gut displacement may not be the best, but we wanted to convey that is a part of the gut, and that the pellets have been displaced, as explained in the discussion.

We have changed to: right and left side of the carapace. This change has been implemented in all figures.

Figure 3. What is the black blob at the anterior end? Possible is better than putative. The three features in F do not look like cavities. Setae? The black objects might be eggs but the text refers to them as “traces”. D and E are not both wet? The location of F should be noted (with an arrow). Is it necessary to label a feature “indeterminate”?

Regarding the anterior black blob, it most probably represents the anterior part of the animal, compressed.

We have changed “putative” to “possible”.

We have remained cautious on the features in F. These black traces appear in some species from the Marble Canyon locality (eg Aria & Caron, 2017), and their nature is indeed puzzling but they appear to be internal.

As noted, D and E are wet

We have added a box to associate the features.

We do not have any suggestion for the nature of the feature labelled as “indeterminate” but still wanted to draw attention to it, in case a reader might have an idea of what this could be.

Figure 4. Lateral views. B is superfluous, and maybe C is not useful. “Part and counterpart overlap” means what? Which part is shown in H?

We have added “lateral views”

We have replaced image in C by an image of another specimen showing a very nicely preserved telson and caudal rami.

By part and counterpart overlap we mean that the image is a composite, with both part and counterpart shown simultaneously, so that the total preservation of the gut can be seen.

In H, the counterpart is shown. We have deleted this image, given its similarities to G.

Figure 6. The letters are too small. Need C and D. The photographic images did not specifically show the dorsal ridge or crest. The caption is too brief. Maybe a ventral view too.

We have updated this figure pointing out different morphological features. Please refer to our interpretation of the carapace “folds” for the presence of the crest.

This is a technical drawing which in many ways is self explanatory.

Adding a ventral view would have been nice but we think the current drawings only convey all the important morphological aspects of the anatomy of this organism.

Figure 7 should have been completed before submission.

We have submitted a completed version, this time.

Figure 8. Fibulacaris is considered closely related to Nereocaris but on the basis of what evidence? The caption is too brief. It should also explain the colors etc. What group is Malacrustacea? This is not a taxon in common usage.

The phylogenetic analysis recovers *Fibulacaris* closely related to *Nereocaris*. We acknowledge that the support values for this association are low. However, *Fibulacaris* shares more common morphological traits with *Nereocaris*, than an alternate close relation to *Perspiscaris* and *Clypeocaris*, as expressed in the discussion.

We have expanded the figure caption accordingly.

“Malacrustacea” is a typo and we refer to “Malacostraca”. We thank the reviewer for spotting this mistake.

Figure 9 should just show Fibulacaris. Page 12 refers to Marrella but this is not relevant and the comparison vague.

We have changed this figure to Supplementary Figure 1 and explained some similarities between *Fibulacaris* and *Marrella* in the Figure’s captions.

Reviewer 2

While the authors are very careful in their conclusions in the main text, making common use of ‘potential.ly’, ‘possible.bly’, etc. their title and abstract are not, and I would suggest them to keep careful therein as well. I am here particularly referring to the title and the use of ‘confirms ’in the last sentence of the abstract, while the authors are way more careful in their discussion

This concern was also brought up by Reviewer 1. We have changed the use of confirm for “suggest” and “may have”. We are aware that ecological reconstructions for fossil taxa are always to a certain degree, speculative, which also extend to the invert-swimming behaviour of *F. nereidis*. As explained by the reviewer, we have been careful in the discussion and conclusions of the text, using extant analogues and functional morphology to state our case. We have rephrased the title accordingly.

In the title, as well as in many places along the text, the authors refer to the long ventral spine of their fossil as a ‘ventro-posterior spine’(even ‘posterior-ventral spine ’p.12 l.43), but this spine is NOT ventro-posterior because it is actually positioned at the anterior end of the animal but is directed towards the posterior. I would rather use ‘posteriorly-directed ventral spine’.

We have changed to postero-ventrally recurved rostrum

There is a big mess with references numbering (a bit detailed below), which absolutely needs to be carefully checked before publication.

We have identified the problem with the references, and we thank the reviewer for spotting this important mistake. We have curated again our bibliography and hopefully this time all concerns have been addressed. We have similarly changed or added references when suggested. Therefore, we consider all the subsequent comments on references addressed (ie. P.3.1.37; p.7 l. 18...).

As the fossils are quite small, the authors state that some anatomical features such as the limbs are poorly preserved or at least hard to observe. This is even more the case in some of the photographs, and I deplore the lack of drawings to help the reader identifying some of the anatomical features described. Similarly, (except for fig. 9) close-ups are not associated with

boxes in the corresponding large-scale photographs and as such are often hard to locate...

4-We have updated our figures with boxes associating close-ups with large-scale photographs. For the suggestion regarding drawings, we have added a drawing of the holotype specimen (Fig. 1B), also please see our response in the comments about figures.

Authors for the taxa are not given. Please add them when you first mention a taxon.

This is not necessary and is not a common practice in this journal.

The phylogeny is probably the main issue of this paper. Besides the fact that Aria & Caron's phylogenies (on which this work is based) are far from making consensus among the community, I noticed the following issues with the way the analysis was performed and the data are presented:

The phylogeny of Aria & Caron has been published before (Aria & Caron, 2017; Vannier et al., 2019, Moysiuk & Caron, 2019, Aria & Caron, 2019), and the dataset represents one of the newest and most comprehensive morphological datasets of Cambrian arthropods up to date. In any case, phylogenies remain hypotheses that are continuously in the process of being falsified by new evidence. The fact that the dataset might not make a consensus among the community may actually be a good thing and we have not seen any papers criticising the aforementioned methods or analyses in detail. As explained in response to reviewer 1, we also do not regard our phylogeny as a critical element of our manuscript, but rather as a useful tool to try to put this species into a broader evolutionary context.

The authors only present a simplified version of a consensus tree from a Bayesian analysis (plus one tree as suppl. fig.), but totally discard parsimony analyses, which in my opinion should be also produced at least for comparison.

We performed a parsimony analysis. However, the number of trees obtained was high, and the consensus tree did not provide any useful information, as polytomies were frequent. In any case, we regard Bayesian analysis as our preferred phylogenetic method. Bayesian has been found to present higher accuracy in morphological datasets (O'Reilly et al., 2016; Puttick et al., 2016) and has been recommended for discrete morphological datasets (Wright & Hillis, 2014; O'Reilly et al., 2018). We prefer to follow some of these analyses, although we acknowledge that phylogenetic methods are still in debate.

(2) The 'simplified version of a consensus tree' is actually NOT a simplified version, as it shows relationships different than in the complete tree presented as suppl. figure; for instance, radiodonts appear monophyletic in Fig. 8, but are absolutely not in the complete tree is suppl. Fig., which is another serious issue...

We acknowledge that the simplification could have obscured some relationships and we have worked upon it. However, for Figure 8, we want the reader to focus on the position of *Fibulacaris* within Mandibulata and therefore leave our tree as simplified as possible. Arthropod workers are encouraged to use the Supplementary Figure 3 to assess the full analysis.

Each new paper using Aria & Caron's phylogeny show important topological changes among the trees, questioning its stability. Here, there is also notable differences with the tree published by Vannier et al. 2018, on which the present analysis is largely based (in particular the position of Hymenocarina and Myriapoda). This should be discussed much more than the couple of sentences

p.9 l.41-48. And the new topology should also be compared to those in both papers published by Aria & Caron in 2017 (your refs [4, 5]).

4) Each new iteration of Aria & Caron's phylogeny has had important changes on the dataset, either adding, deleting or redefining characters. Similarly, the number and species included has changed from Aria & Caron 2017, to Vannier et al., 2019 to the present analysis. Furthermore, phylogenies on Cambrian arthropods suffer from a less than ideal dataset in the number of characters and species and have been hindered by discussions on homologies and therefore are more liable than extant phylogenies (Aria & Caron, 2017). We added some sentences reflecting these changes, but a full analysis on types of analyses and dataset is beyond the scope of this project.

(4) The 'four new characters' added (p.5 l.23) should be described in the main text, and the effect of their addition at least discussed (showing trees with and without these characters, at least in Suppl. Material, would be very useful). Note that they are not much discussed in Suppl. Material either.

The new characters can be found in the Supplementary Material 3 and we have added some sentences regarding their influence on the phylogeny in the Results section. We have similarly added a tree without the new characters (Supplementary Figure 2).

(5) The backbone constraint used (p.5 l-3) can only be found in the matrix nexus file, but nowhere else in the main text or suppl. material.

5) We have added the backbone constrain in the Supplementary Material, as suggested.

Fig. 3A, the long posteriorly-directed ventral spine seems segmented/annulated, is this due to the quality of the photograph in the pdf version, or could it be a real feature?

We have rechecked our fossil specimens, and there is no evidence of segmentation in the spine. Segmented structures, such as antennae, are preserved more clearly in taxa of similar sizes (e.g. *Perspicaris dictynna*) and therefore we regard any segmentation in the spine as speculative.

And a totally open question: Considering that *Fibulacaris* specimens are quite small (maximum length: 2 cm) compared to most other bivalved arthropods from the Burgess Shale, and considering also that conspicuous spines are common in larval/juvenile crustacean stages, couldn't *Fibulacaris* represent a larval/juvenile form of a larger bivalved arthropod found in the same layers? Of course, the putative eggs (an unlikely interpretation in my opinion) would make them mature individuals, but this constitutes a weak (and very unlikely) argument to rule out the possibility that they were larval/juvenile forms.

We do not consider our fossils to represent larval stages although the smallest individuals (ca. 4mm) could represent juvenile forms. In the Marble Canyon and Tokumm communities, most bivalved arthropods bigger than *F. nereidis* are *Tuzoia*, *Tokummia* and *Canadaspis* (Caron et al., 2014). These species are morphologically very different from *Fibulacaris* (e.g. Presence of frontal appendages and mandibles in *Canadaspis* and *Tokummia*, limb structure, presence of the spine in *F. nereidis*, morphology of the telson...). Furthermore, none of the limbs in *F. nereidis* appear to be buds, which may indicate a larval affinity.

We acknowledge that the putative eggs are tentative, and we do not use this as an evidence for *F. nereidis* being an adult stage.

Reviewer 2. Additional comments

- 2.1.27: We have extended our comparison with trilobite structures in the discussion.
- 2.1.33: We have added this suggestion to the abstract
- 3.1.7: Added extant taxa as examples
- 3.1.14: Reshaped this section and added references to fossil taxa as well.
- 3.1.36. -3.1.37. -3.1.39: References have been checked through the text. Please note that the reduction of the introduction deleted some of the previous references.
- 3.1.39: Changed to “as well as”
- 4.1.16: Changed to “(ROMIP) collections”
- 4.1.53: Figure 9 was changed to Supplementary Figure 1.
- 5.1.13: Figure 9 was changed to Supplementary Figure 1.
- 5.1.13: We agree with the reviewer that the carbon is possibly just topographic and have changed the sentence. Also changed it to phosphorous.
- 5.1.14: We refer to previous work on the subject. Please note that is not the focus of this paper to discuss the complex process of mineralization.
- 5.1.20-21: We are referring to Vanniet et al., 2019. This, and the number of characters (209 from their dataset) have been amended.
- 5.1.23-24: Changed to just “multistate”
- 5.1.41: Corrected to von Siebold.
- 5.1.49: We do not think that the latin name has to necessary express the affinity. “Caris” represents a broad term, which is used frequently in Cambrian arthropods, even recently described species (*Jugatacaris*, *Erjiecaris*, *Surusicaris*).
- 6.1.3/6.1.11-12: We have reworked on the diagnosis, following the suggestions of Reviewer 1.
- 6.1.8: In this case we mean that the carapace height is bigger than the body height, therefore leaving the afore-mentioned wide space between the body and the upper part of the carapace
- 6.1.14: Changed to “*Fibucalaris nereidis* sp. nov”
- 6.1.19-21: We welcome the suggestion, but we do not think that all specimens figured should be considered paratypes. Some specimens figured were chosen for very specific features (eg. Figure 4B) but are not some of the best preserved among the figured ones and may lack key diagnostic features.
- 6.1.30: We have indicated the specimen numbers, as suggested. In both cases, the measurements are 20.0 and 10.0 mm.
- 6.1.36: We agree with the reviewer and have included Figure 1, as well.
- 6.1.45: We agree with the reviewer and have changed to “less than one third”
- 7.1.45: Sadly, the number of specimens is too small to infer the presence of morphotypes. We have added a supplementary figure (Supplementary Figure 5) that showcases the length and width of all specimens measured.

-7.1.15: We have added the carapace folds in the figures as suggested. Please note that the names may have changed following the suggestion of Reviewer 1.

-7.1.18. References changed

-7.1.23: By “differential position of the legs” we mean that legs sometimes appear touching the spine, while in other cases only reach the edges of the carapace. We have changed the sentence to make the concept more explicit.

-7.1.41/8.1.7: Changed

-7.1.47: We acknowledge the problem and have deleted the words “closely related”

-8.1.30: Changed to “Figure 2F”

-8.1.41: We have added a new figure (Now Figure 4 C) that shows a better close-up of the telson.

-8.1.44: We have followed the reviewer’s suggestion and added Figure 3D to the referred figures.

-8.1.46: We have entirely replaced Figure 4 C.

-8.1.50: The traces are spherical (ie. 3D), we have changed to sentence to make it more explicit.

-9.1.21: Changed to *Fibulacaris nereidis* gen. et sp. nov.

-9.1.31: We refer to frontal processes as any frontal structure that is an extension of the carapace, and which can be potentially analogous to the spine in *F. nereidis*. We have changed the sentence for clarification.

-9.1.44-45: We refer now to Figure 8, as suggested and we have made sure to refer more often to the figures.

-10.1.29: Changed to “rostrum”.

-10.1.42-45 We do not actually have any specific hypothesis for this tissue, although a similar cavity to that of *Marrella*’s spines could be possible (Supplementary Figure 1). Nervous tissues in the Burgess Shale tend to be preserved in Carbon (see Parry and Caron 2019; Aria and Caron, 2017 as examples)

-11.1.3. -11.1.21-11.1.45 References changed

-11.1.53: Taxa ordered as suggested

-11.1.56: Reference rechecked

-12.1.6: The sentence widely changed

-12.1.17: Added several references pointing out invert swimming behaviours

-12.1.18-19. Reference changed

-12.1.23-24. Reference added

-12.1.33: Added “are”.

-12.1.37: Overall reference curated

-12.1.43: Changed to “rostrum”, following Reviewer 1 comments.

- 12.1.44: We changed the sentence to “crustaceans or other arthropods”
- 12.1.49: Please, see our response in Major comments 1
- 13.1.6-7: Changed as suggested
- 13.1.7: The sentence was rewritten.
- 13.1.12-18: We agree with the reviewer and have completely reworked this section appropriately.
- References curated as suggested

Figures

Fig. 1, legend mouth doesn't show the same structure in A, B and E, could you please be more precise. Also, scale bar in C is barely visible, could you perhaps use a white colour instead of black? White scale bars should probably be used for each photograph.

The mouth can only be inferred through the first phosphatized segment. We have moved the position of this feature across different figures to make it more constant. Please note that Figures 1B, C, D were replaced.

Scale bar was changed to white when the background was shown too dark.

Fig. 2 caption, specimen shown in C is indicated as the counterpart of specimen in A, but they are clearly the same. C is just a close-up of A, isn't it? Explain also, as you do in Fig. 3 caption, why the eye is visible in C but not A ('different polarization'). Also, a box locating the area shown in F is needed on photograph E.

B and C are the counterpart of A, please note how the eye on the spine notch is not visible in A.

C is a closeup of B. We have added it into the captions for further clarification.

All images were taken with the same polarization.

We have followed the suggestion of the reviewer and added a box.

Please note that we replaced Figure C and D with other specimens.

Fig. 3, scale bars in A, C, D, F and G should be white. Also, line drawings of the telson would be useful as it appears very dark and the caudal rami (c.r.) is particularly hard to see in the photographs (see above). More importantly, there is a clear labelling problem in valves orientation between photograph A and B, where the same valve is labelled l.v in A and r.v. in B. Considering that you have a ventral view (and I concur with this interpretation as one would expect to see the hinge in dorsal view, which is clearly not the case here, even though you needed to prepare out the spine on one side, which was very likely because dorso-ventral flattening of this specimen resulted in the spine to be squished between the valves) then you should look at the part, and the valve appearing on top of photograph A is indeed the l.v, therefore so should also be the one on top of photograph B. Finally, boxes locating the close-ups shown in C–F are needed on photograph B, absolutely required for F.

Scale bars were changed where the background was too dark.

We appreciate the suggestion of the reviewers, but Figure 3 C gives a clear view of the morphology of the caudal rami, which is complemented by a new figure (Figure 4C).

We agree with the reviewer and have changed the names of the valves accordingly, which are now referred as right or left side of the carapace.

Boxes have been added as suggested.

Fig. 4 caption, the use of 'specimens with 'highly 'preserved guts 'is unclear to me. Don't you actually mean 'specimens with 'the best 'preserved guts'?

We changed the term following the reviewer's suggestion, although the full caption has been changed given the new specimens presented.

Fig, 5, scale bar in D should be white. Also, a box locating the area shown in H is needed on photograph F. Fig. 5 caption, based on the apparent symmetry between specimens in E and G, it looks like the specimen shown in G. should be ROMIP 65379 'counterpart'. Also, abbreviation le) is indicated as to be 'leg 'here, while you indicate it as 'legs 'in other captions, and you obviously show also several here. Close-ups on the c.f. in I and s.f. in K are needed as they are otherwise absolutely not visible.

Scale bars were changed to white where the background was too dark.

Boxes have been added as suggested.

We agree with the reviewer and added "counterpart"

We have changed it to "legs" as suggested.

Please note the black lines on some of these specimens, specially Figure 5K and 5I

Fig. 6 lacks letters accompanying the different diagrams, especially as the letters used in the caption do not follow the order of the diagrams in the figure. I would put the frontal view first (A), then lateral reconstruction (B), cross section through the cephalic region (c) and finally cross-section through the thorax (D). And don't forget to label the lines locating the crosssections in the lateral view as C and D. Fig. 6 caption, A and B (or let's say the first to left diagrams in Fig. 6) are cross-sections across the cephalic and thoracic regions respectively. C (the third diagram from the left) is the frontview. Please correct that.

We followed the reviewers' suggestions and added notes on several morphological features of the diagrams.

We changed the order of the diagrams and the captions as suggested.

Fig. 7 caption, it may be worth indicating that you represented *Fibulacaris* during inverted swimming.

We have changed the figures captions accordingly

Fig. 8, the use of a silhouette of *Anomalocaris* in front of the Isoxyiids is misleading.

Also, the blue dot with the silhouette of *Fibulacaris* appearing in front of *Nereocaris briggsi* is a bit misleading to the reader.

We have changed the place of *Anomalocaris* as suggested

We have similarly deleted the blue dot

Please note that the analysis was re-run, eliminating one character (presence of mouth-which was uninformative).

Fig. 8 caption, see above but this tree is clearly not a 'simplified version' of the tree shown in suppl. fig. but more an 'interpretative tree based on your consensus tree' Please rephrase the title.

We have amended some of the tree branches and changed the caption to 'interpretative tree based on the consensus tree' as suggested.

Fig. 9 caption, D) caption shouldn't be in bold. And phosphate in D) and E) should be 'phosphorous' as you refer here to the element, not the ion or compound. Also, you say that the matrix comprises 214 characters, but there are only 213 in suppl- material (see above), but indeed.

All bold sections of the captions were replaced.

Changed to phosphorous as suggested. Please note that Figure 9 was changed to Supplementary Figure 1.

The number of characters has been changed to 213, and the phylogenies have been re-done accordingly